# A postmeiotically bifurcated roadmap of honeybee spermatogenesis marked by phylogenetically restricted genes

**Zhiyong Yin**[1☯], **Guiling Ding**[2,3☯], **Yingdi Xue**[1], **Xianghui Yu**[1], **Jie Dong**[4], **Jiaxing Huang**[2,3]*, **Jun Ma**[1,5,6,7]*, **Feng He**[1,5,6,7]*

1 Center for Genetic Medicine, the Fourth Affiliated Hospital, Zhejiang University School of Medicine, Hangzhou, Zhejiang, China, 2 State Key Laboratory of Resource Insects, Institute of Apicultural Research, Chinese Academy of Agricultural Sciences, Beijing, China, 3 Key Laboratory for Insect-Pollinator Biology of the Ministry of Agriculture and Rural Affairs, Institute of Apicultural Research, Chinese Academy of Agricultural Sciences, Beijing, China, 4 Institute of Animal Husbandry and Veterinary Science, Zhejiang Academy of Agricultural Sciences, Hangzhou, China, 5 Women's Hospital, Zhejiang University School of Medicine, Hangzhou, Zhejiang, China, 6 Institute of Genetics, Zhejiang University International School of Medicine, Hangzhou, Zhejiang, China, 7 Zhejiang Provincial Key Laboratory of Genetic and Developmental Disorder, Hangzhou, Zhejiang, China

☯ These authors contributed equally to this work.
* huangjiaxing@caas.cn (J.H); jun_ma@zju.edu.cn (J.M); feng_he@zju.edu.cn (F.H)

**Data Availability Statement:** The authors declare that all data supporting the findings of this study are available within the paper and its supplementary information. All raw deep

## Abstract

Haploid males of hymenopteran species produce gametes through an abortive meiosis I followed by meiosis II that can either be symmetric or asymmetric in different species. Thus, one spermatocyte could give rise to two spermatids with either equal or unequal amounts of cytoplasm. It is currently unknown what molecular features accompany these postmeiotic sperm cells especially in species with asymmetric meiosis II such as bees. Here we present testis single-cell RNA sequencing datasets from the honeybee (*Apis mellifera*) drones of 3 and 14 days after emergence (3d and 14d). We show that, while 3d testes exhibit active, ongoing spermatogenesis, 14d testes only have late-stage spermatids. We identify a postmeiotic bifurcation in the transcriptional roadmap during spermatogenesis, with cells progressing toward the annotated spermatids (SPT) and small spermatids (sSPT), respectively. Despite an overall similarity in their transcriptomic profiles, sSPTs express the fewest genes and the least RNA content among all the sperm cell types. Intriguingly, sSPTs exhibit a relatively high expression level for Hymenoptera-restricted genes and a high mutation load, suggesting that the special meiosis II during spermatogenesis in the honeybee is accompanied by phylogenetically young gene activities.

## Author summary

Previous histological studies suggest that spermatogenesis is concluded before adult emergence in the honeybee. We complement this histological atlas on testis development and spermatogenesis with a comprehensive dataset from the honeybee adult testes, including single-cell RNA, long-read RNA, bulk mRNA and DNA-seq data. Through the use of

sequencing data have been deposited at SRA under
BioProject: PRJNA981429.

**Funding:** FH was supported by two National Key
R&D Program of China grants (2018YFA0800102
and 2021YFC2700403) and a National Natural
Science Foundation of China grant (31871249). JM
was supported by a National Natural Science
Foundation of China grant (31871452). JH was
supported by a National Key R&D Program of
China grant (2022YFD1600205), the China
Agriculture Research System-Bee (NYCYTX-44-
KXJ5) and the Agricultural Science and Technology
Innovation Program Chinese Academy of
Agricultural Sciences (CAAS-ASTIP-2023-IAR).
The funders had no role in study design, data
collection and analysis, decision to publish, or
preparation of the manuscript.

**Competing interests:** The authors have declared
that no competing interests exist.

gene orthology between the honeybee (*Apis mellifera*) and *Drosophila melanogaster*, we
show that in newly emerged drones, spermatogenesis is an ongoing process and meiosis II
results in two separable clusters of spermatid cells. One of them is identified to contain
small spermatids, which are consistent with previous anatomical detection of polar body-
like cells in close association with regular spermatids. The characteristic features of small
spermatids include an overall low transcription level, high expression of evolutionarily
young genes, an enrichment of gene activities regulated by ETS-domain transcription fac-
tors, and a high level of mutation load. This work establishes the foundation for future
investigations into molecular mechanisms of sperm quality assurance in the honeybee.

## Introduction

A central question in developmental biology is to what extent deviations from the general par-
adigm of reproduction occur across the animal kingdom. An illustrative example comes with
the haplodiploid sex determination of hymenopterans such as ants, bees and wasps, in which
unfertilized and fertilized eggs develop into males and females, respectively [1,2]. But how
these haploid males with only a single set of chromosomes undergo meiosis and produce sper-
matozoa remains elusive. In hymenopteran species, each spermatocyte undergoes no nuclear
division during a presumed meiosis I [3–6]. Such an altered meiotic process skips the step of
reducing genetic material to preserve a relatively conventional meiosis II. With complete
breakdown of nuclear envelope and complete separation of sister chromatids, each late-stage
spermatocyte divides into two haploid spermatids. Interestingly, the hymenopteran male mei-
osis II can also contribute to the diversity of reproductive development. In ants and wasps, the
two spermatids that arise from the same spermatocyte appear indistinguishable [4,5,7]. In con-
trast, in bees (Hymenoptera: Apidae), such as *Apis dorsata*, *Apis mellifera*, *Melipona bicolor*,
*Melipona quadrifaciata*, *Scaptotrigona postica*, and *Xylocopa fenestrata*, the cytoplasmic sepa-
ration during meiosis II is unequal, giving rise to one typical spermatid and one small haploid
cell [4,5,8–10]. While it is well documented that the small cells contain little cytoplasm and few
membranous organelles [4,11,12], it is still subject to debate as to what these small cells are des-
tined to become. There are two long-standing hypotheses derived from microscopic observa-
tions and estimations of the spermatocyte-to-spermatozoon ratio: these small haploid cells
either degenerate or develop into mature sperm cells at a later stage [13–17]. However, there is
currently a complete lack of molecular characterizations of these cells with regard to their
developmental origin or fate.

Existing knowledge about gonad and germ cell development of the honeybee (*Apis melli-
fera*, Amel), which is based on observational studies that date back to the early twentieth cen-
tury, highlights additional differences from other organisms, such as the dipteran fruitfly
(*Drosophila melanogaster*, Dmel), underscoring a need for molecular studies in this organism
that is also of agricultural importance. For example, although the embryonic origin of male
gonads is evident, there is no support for the presence of pole cells in the honeybee embryo
and there is no consensus for the location of male germline stem cell niche [12,18–24]. In the
fruitfly, one testis can be viewed as a single seminiferous tubule, in which each developmental
stage within a germ cell cyst occurs consecutively and there is a gradient of spermatogenesis
progression along the apical-to-basal axis [25,26]. In contrast, a honeybee testis consists of
150~200 seminiferous tubules, in which there exist both within-tubule and between-tubule
variations [3,13,23]. It is generally thought that, unlike the fruitfly males, the honeybee drones
produce virtually all spermatozoa prior to emergence [27–29]. However, limited histological

studies have not fully resolved this issue stemming from a lack of molecular markers for effectively labelling sperm cells of distinct developmental stages.

Despite the differences in testis morphology across different animal species, the core spermatogenic functions are evolutionarily conserved. Such conservation limits the transcriptome divergence in spermatogenic cells [30]. This allows the utilization of marker transcripts with shared orthology to identify major cell types in testis single-cell RNA sequencing (scRNA-seq) datasets across species [30–32]. Here we present single-cell transcriptomic maps of testis samples from honeybee drones at 3 and 14 days after emergence (3d and 14d). Through the use of conserved marker genes that were well-characterized in *Dmel*, we identify major testicular cell types in the adult drones. Our analysis also uncovers a cluster of cells with several key features indicative of small spermatids (sSPT) arising from unequal meiosis II. The distributions of different cell types in 3d and 14d testes provide evidence supportive of the sSPT elimination hypothesis. In addition, our results shed an evolutionary light on a role of Hymenoptera-restricted genes (HRG) in crafting the unique paradigm of male meiosis II in the honeybee.

## Results

### Testis scRNA-seq reveals a continuous process of spermatogenesis in newly-emerged honeybee drones

We collected drones of 1~18 days after emergence from hives that were visually inspected to be healthy (Materials and Methods). We dissected immediately the reproduction system including mucus glands (MG), seminal vesicles (SV), and testes (T; see Fig 1A and 1B for representative images of the dissected organs from 3d and 14d drones, respectively). For each drone, we separately counted the total spermatozoa in the testis pair and the SV pair. A newly emerged drone of 1~2-day old had an average number of 0.95 ± 0.22 million spermatozoa in testes and almost none in SVs. The number of testicular spermatozoa showed a gradual declined until day 10, reaching a base level of 0.04 ± 0.03 million (Fig 1C red). Inversely, the number of SV spermatozoa elevated during days 3~10 (Fig 1C blue). Our measured average number of sperms and their daily changes are comparable to those in previous reports, supportive of normal and healthy development of our collected drones [33–36].

For the current study, we prepared scRNA-seq libraries from freshly dissected testes of 3d and 14 drones, respectively (the samples were collected using the same procedures as described above). The 3d of age represents the onset of the transition stage when a massive number of spermatozoa are transferred from testes to seminal vesicles, while the 14d of age is a time when drones are fully mature for mating flight. After the removal of low-quality cells including damaged cells, small cytoplasmic blebs and doublets, we recovered a total of 10,783 and 6,842 cells of 3d and 14d testes, respectively (S1A–S1E Fig; see also Materials and Methods). Mapped and annotated with the reference genome Amel_HAv3.1, a total of 11,242 and 11,375 genes showed testicular expression under the single-cell resolution (at least one unique molecular identifier detection in at least 3 cells), respectively. Both datasets exhibited a strong correlation (Pearson's $R$ = 0.97 and 0.97, respectively; $p$-value $< 10^{-16}$) with the corresponding whole-testis RNA-seq datasets from drones collected using the same procedures (Materials and Methods), suggesting reliable detection of gene expression.

We performed UMAP clustering with Seurat to group cells by the similarity across their unique gene expression features. By testing different Seurat parameters, we found that 9 unsupervised clusters allowed quality identification of the major testicular cell types in the 3d dataset (Fig 1D; Materials and Methods). To infer the cell identities represented in individual clusters, we relied on the knowledge in *Drosophila* (see S1 Table for a list of 135 *Amel* genes with one-to-one orthology to the testicular-cell-type-specific genes of *Dmel*). Among these

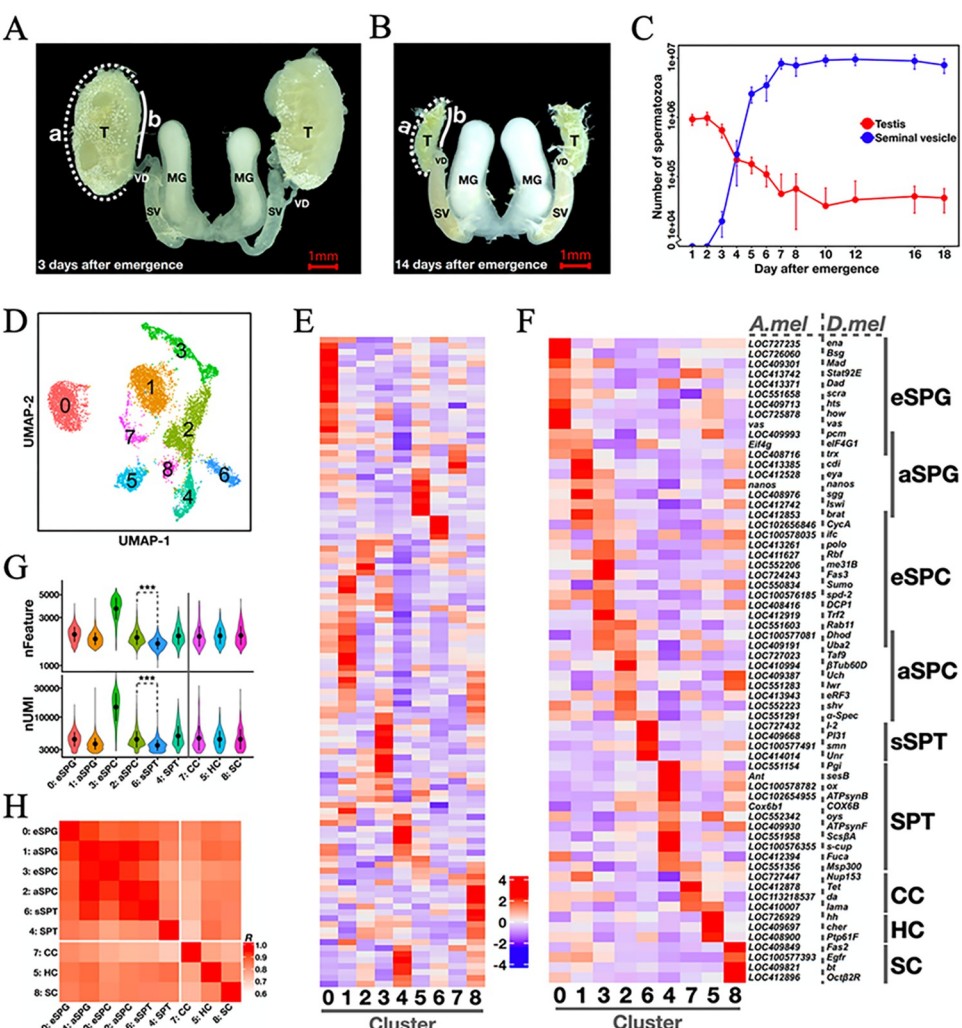

**Fig 1. Spermatogenic cell types are identified by testis scRNA-seq from 3-day-old honeybee drone adults.** (A-B) Testes of healthy drones at 3 and 14 days after emergence. T: testis; MG: mucous gland; SV: seminal vesicle; VD: vas deferens; *a*: the apical side of a testis; *b*: the basal side of a testis. (C) The average numbers of spermatozoa in testes (red) and seminal vesicles (blue) from a healthy drone collected at 1~18 days after emergence. N = 20 for each measurement. Newly emerged drones within the first two days have no detection of spermatozoa in seminal vesicles. For the other data points, errorbars represent one standard deviation. (D) A UMAP projection of 10,783 high-quality cells from 3d testes grouped into 9 unsupervised clusters. (E) A heatmap showing the expression of 109 candidate marker genes that are orthologous to the *Dmel* marker genes. A total of 135 one-to-one orthologous genes were filtered according to two criteria. 1) The gene was expressed in >10% cells of at least one cluster. 2) At least one marked cluster expressed the gene at a level >50% of the maximum level among all clusters. See S1 Table for the complete lists of gene names and the *Dmel* references. (F) A heatmap showing the expression of 64 honeybee marker genes that exhibit reliable cell-type-specific expression in 3d testes. eSPG: early spermatogonia; aSPG: advanced/late spermatogonia; eSPC: early spermatocytes; aSPC: advanced/late spermatocytes; SPT: spermatids; sSPT: small spermatids; CC: cyst cells; HC: hub cells; SC: sheath cells. (G) The numbers of genes (nFeature) and unique RNA molecules (nUMI) that were detected in each cluster. The average RNA molecules in sSPT (nUMI = 3,642 ± 826) were significantly lower than those in aSPC (4,705 ± 1,862; Student's t-test $p < 10^{-4}$) or those in SPT (5,461 ± 2,413; $p < 10^{-4}$). (H) Pearson's correlation coefficients between each pair of cell types identified from 3d adult testes.

genes, 109 had enriched expression in at least one single-cell cluster of the 3d dataset (Fig 1E; Materials and Methods). Based on these genes, we annotated each cluster with the *Dmel* cell type information, iteratively removed genes with either inconsistency or high ambiguity, and obtained a final set of 64 marker genes (Fig 1F; Materials and Methods). Using these marker

genes, we identified three testicular somatic cell types and six spermatogenic cell types: cyst cells (CC), hub cells (HC), sheath cells (SC), early spermatogonia (eSPG), advanced/late spermatogonia (aSPG), early spermatocytes (eSPC), advanced/late spermatocytes (aSPC), spermatids (SPT), and small spermatids (sSPT). We designated cells in the identified sSPT cluster as small spermatids with following considerations. First, both the read quality and the single cell quality in the sSPT cluster were as high as the other testicular cell types in our honeybee data and in published fruitfly data (S1F–S1G Fig). Second, while the presence of sSPT cluster was robust in the unsupervised clustering analysis against the threshold choices for transcriptional activity (nFeature and nUMI), these cells had the lowest overall level of transcription (nFeature = 1,667 ± 227 and nUMI = 3,642 ± 826) among all the cell types (Fig 1G). Third, the sSPT cluster adjoined to aSPC and SPT on UMAP (Fig 1D), and appeared as a branch in between aSPC and SPT along the pseudotime axis (see below). Fourth, the transcriptomic features of sSPTs were biased toward germ cell types rather than somatic cell types (Fig 1H). In fact, the sSPT cluster was correlated the best with aSPCs (Pearson's $R = 0.94$), favoring the hypothesis of sSPTs being derived from late spermatocytes. Importantly, we were compelled by the evidence that the identified sSPTs were characterized by an enriched expression of Hymenoptera-restricted genes (HRGs) with their expression levels reaching or surpassing those of other genes (see below). This result ruled against the possibility that the identified sSPTs were merely cytoplasmic fragments derived from other cells. Together, these considerations led us to suggest that this sSPT cluster likely represented the post-meiotic, small spermatid cells, a type of cells previously reported in anatomic studies [11]. Furthermore, the spermatogenic cell clusters, including sSPT, were well correlated with one another (Pearson's $R > 0.77$; Fig 1H), suggesting a continuous process of spermatogenesis presented in the newly-emerged honeybee testes.

## Expression and evolution signatures of the spermatogenic cell type-specific marker genes

The expression patterns of the 53 *Amel* spermatogenic marker genes used in annotating the 6 spermatogenic types of cells in 3d honeybee testes (Fig 1F) were consistent with the knowledge of their *Dmel* counterparts (see S1 Table for a complete list of references). For instance, *Dmel\how* encodes an RNA-binding protein required for mitotic progression of germline stem cells (GSC) and gonialblasts [37]. Its *Amel* ortholog (*Amel\LOC725878*) was also biased toward the early stage of spermatogenesis and could serve as a specific eSPG marker. Some of the marker genes, such as *Amel\LOC409993* (*Dmel\pcm*), *Amel\Eif4g* (*Dmel\eIF4G1*) and *Amel\LOC408716* (*Dmel\trx*), had a relatively high expression in both eSPG and aSPG clusters. A better aSPG marker gene is *Amel\LOC413385*, which had much lower expression in all the other spermatogenic clusters. Its *Dmel* ortholog (*Dmel\cdi*) encodes a testis-specific serine/threonine kinase and also shows specific expression in dividing spermatogonia [38]. Among the 21 genes used to annotate the two clusters of spermatocytes, *Amel\LOC412919* was exclusively enriched in the eSPC cluster, consistent with the observation that the expression of the *Dmel* ortholog (*Dmel\Trf2*) during spermatogenesis coincides with initiation of the meiotic program [39]. Interestingly, a similar correlation has also been reported between the onset of meiosis and the onset of *Dmel\βTub60D* expression [40], but the expression of its *Amel* ortholog, *Amel\LOC410994*, had a strong peak of expression in the aSPC cluster. For the two spermatid clusters, SPT and sSPT, *Amel\LOC551958* and *Amel\LOC409668* emerge as the specific marker genes, respectively. Their corresponding orthologs, *Dmel\ScsβA* and *Dmel\PI31*, both function in caspase activation during sperm individualization [41,42].

To evaluate the expression evolution of these honeybee spermatogenic marker genes, we built an expression tree for these genes based on bulk RNA-seq data from adult testes of *Dmel*,

*Amel* and three other Hymenopteran species: *Bombus terrestris* (*Bter*), *Nasonia vitripennis* (*Nvit*) and *Solenopsis invicta* (*Sinv*). This tree recapitulates the known Hymenopteran phylogeny (S2A Fig), suggesting that the expression changes of these genes steadily accumulated within Hymenoptera. Indeed, these genes show significantly higher testis-bulk expression on average than the other genes in all the five species (S2B Fig). To further examine the evolution signatures of these genes, we estimated their $d_S$, $d_N$ and $d_N/d_S$ using codeml model 0 [30,43–45] across the genus of *Drosophila* or *Apis* (Materials and Methods). For both synonymous ($d_S$) and nonsynonymous ($d_N$) substitutions, the rates in *Apis* were significantly smaller than those in *Drosophila* (Wilcoxon rank sum test $p = 10^{-25}$ and $10^{-11}$, respectively; S2C Fig). However, the $d_N/d_S$ ratios were similar ($d_N/d_S = 0.08 \pm 0.08$ and $0.06 \pm 0.06$, respectively; Wilcoxon rank sum test $p = 0.80$; S2D Fig). As a result, although there is an overall difference in evolution rates between the two genera, the spermatogenic marker genes identified from our honeybee scRNA-seq analysis are constrained by strong purifying selection in both *Apis* and *Drosophila*.

## The earliest spermatogonial cells express germline stem cell maintenance genes

It is generally believed that spermatogenesis is completed by the end of the pharate-adult stage. In the daily measurements of the sperm numbers, we detected a non-significant increase of the testicular spermatozoa at day 2 after emergence (from $0.9 \pm 0.2$ million to $1.0 \pm 0.2$ million, Student's t-test $p = 0.23$; Fig 1C red), indicative of presumed new sperm formation. During days 3–8, the gain rate in SVs was significantly greater than the loss rate in testes (Student's t-test $p < 10^{-7}$ based on bootstrapping distributions; S3 Fig), suggesting an unremitting process of spermiogenesis as well as a shrinkage of the spermatogonium pool. In addition, by performing Hematoxylin and Eosin (H&E) staining on testicular sections, we confirmed that both spermatogenic cysts and spermiogenic bundles are simultaneously present in the same seminiferous tubules of the 3d adult testis (Fig 2A; see S4 Fig for more representative sections). Within each tubule, the bundles of elongating spermatids were located near the central axis with their sperm heads oriented towards the basal side (Fig 2A arrowheads). More peripherally, the round-shaped germ cell cysts scattered without any obvious difference in either the density or the size along the apical-basal axis (Fig 2A dashed circle). These results show that, contrary to the popular belief, male germ cells of early spermatogenic stages are present in the testes of newly-emerged honeybee drones.

In our scRNA-seq dataset of 3d adult testes, the two spermatogonial clusters consisted a total of 4,776 cells (Fig 1D red and yellow). The pseudotime trajectory analysis of these cells yielded a developmental continuum from the beginning of eSPG to the end of aSPG with no branch point (Fig 2B). Hierarchical clustering of a total of 4,271 genes that were differentially expressed between eSPG and aSPG identified two major waves of gene activation (Groups #2 and #3 in Fig 2C). Group #2 consisted of 2,077 genes, which display earliest spermatogonia-specific gene expression and top enrichment in functional categories such as cell differentiation (GO: 0030154), gamete generation (GO: 0007276) and negative regulation of DNA-templated transcription (GO: 0045892; all *Amel* GO analyses were performed by the tool of HymenopteraMine, and the adjusted $p$-values were obtained by hypergeometric test with Holm-Bonferroni correction; see S2 Table for the complete results). Group #3 consisted of 1,432 genes, showing transcriptional activation in the latest spermatogonia with enrichment in cell differentiation (GO: 0030154), positive regulation of DNA-templated transcription (GO: 0045893), Hippo signaling pathway (KEGG: ame04391), and WNT signaling pathway (KEGG: ame04310). While cells near the end-point along the spermatogonial pseudotime axis likely represent the cellular process of spermatogonium-to-spermatocyte transition, the earliest cells

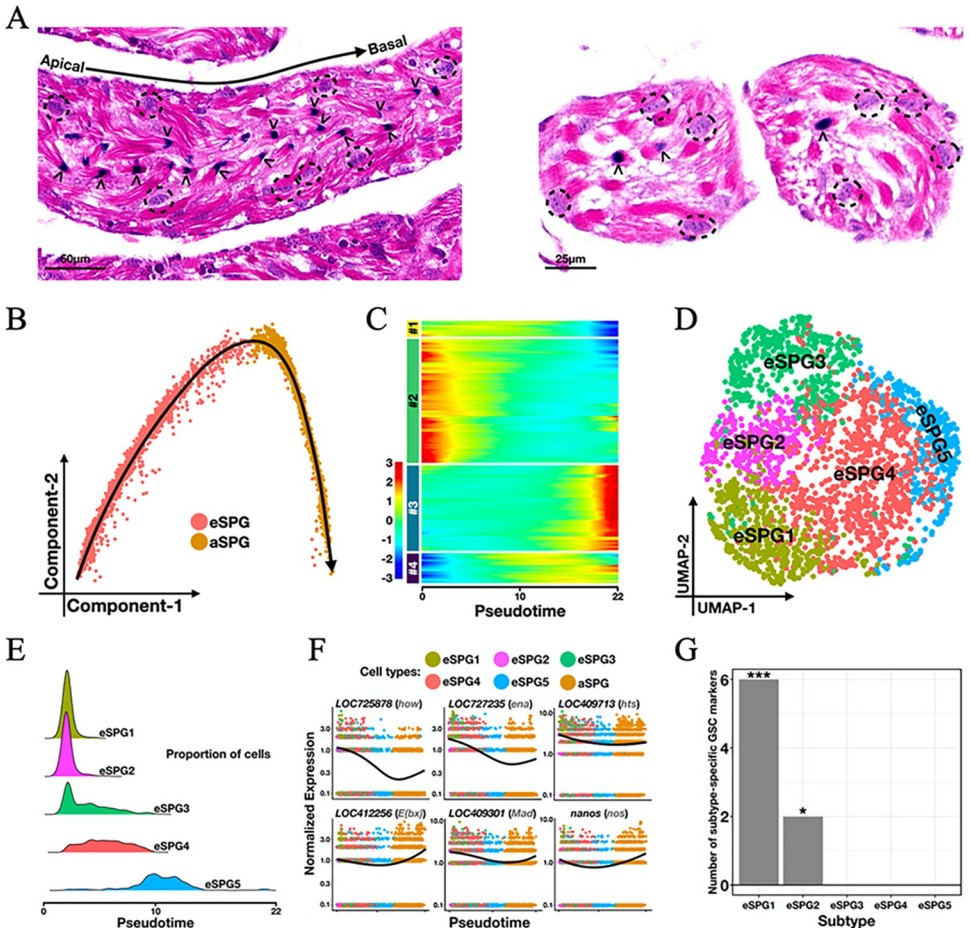

**Fig 2. Germline stem cell markers are expressed in the earliest subpopulation of spermatogonial cells from 3-day-old honeybee drone adults.** (A) Midsagittal (left) and cross (right) sections of seminiferous tubules from 3d honeybee adult testes. Hematoxylin and Eosin staining were used. Dashed circles: spermatogenic cysts; arrowheads: spermiogenic bundles. (B) Spermatogonial cells ordered along pseudotime. Red: 2,460 eSPG cells; yellow: 2,316 aSPG cells. (C) A pseudotime-ordered heatmap showing the dynamic expression of 4,271 differentially expressed genes (adjusted $p$-value $< 0.001$ calculated by Seurat) in spermatogonial cells. Hierarchical clustering reveals four major gene groups with distinct patterns. (D) UMAP clustering of the early spermatogonial subset. (E) Distributions of each early spermatogonial subcluster on the pseudotime axis. The trajectory in panel B was used. (F) Dynamic expression of six germline stem cell (GSC) marker genes that are differentially expressed (DEGs) in eSPG1. GSC markers were determined according to the *Amel*-to-*Dmel* orthology. DEG analysis was performed using Seurat. Colored dots: individual cells from different eSPG subclusters; solid lines: loess-smoothed mean profiles along the pseudotime trajectory in panel B. (G) 6 and 2 GSC marker genes were differentially expressed in eSPG1 and eSPG2, respectively (hypergeometric test $p = 10^{-6}$ and 0.02, respectively). No GSC marker gene was determined as DEGs in the other eSPG subclusters.

may contain undifferentiated spermatogonia or germline stem cells (GSC) due to high similarity in transcriptome.

To further characterize the early spermatogonial cells, we re-clustered the 2,460 eSPG cells (Fig 2D). This resulted in five subclusters, eSPG1 to eSPG5, with distinct distributions along the pseudotime axis (Fig 2E). Differentially expressed gene (DEG) analysis showed that the eSPG1 cells expressed six genes that share 1-to-1 orthology with known GSC markers in *Dmel*, including *nanos*, *LOC409301* (*Dmel\Mad*), *LOC409713* (*Dmel\hts*), *LOC412256* (*Dmel\E(bx)*), *LOC725878* (*Dmel\how*) and *LOC727235* (*Dmel\ena*) [25,26,37,46–48]. The expression of these genes peaked in eSPG1 and declines as the pseudotime goes (Fig 2F). In contrast, only

*nanos* and *ena* were enriched as eSPG2 marker genes, and none for the other three subtypes (Fig 2G). These results suggest that the 3-day-old adult testes of honeybee harbor the earliest SPGs with a substantial differentiation potential, in addition to the more developed SPGs with a minimal differentiation potential.

## Spermatogenesis in honeybee is characterized by a branch point after meiosis II

For mouse and fly, the pseudotime profiles of single meiotic cells usually show a developmental continuum [49,50]. Here we excluded somatic cells and spermatogonia from the 3d data, and performed pseudotime inference on a total of 4,550 peri-meiotic cells. The resulting trajectory revealed a notable branch point (Fig 3A). Following this point, the sperm cells were bifurcated into two separate lineages, one having 60.3% SPT cells and the other having 85.9% sSPT cells. While the eSPC-aSPC-SPT trajectory resembled the developmental continuum seen in mouse and fly, the aSPC-sSPT branch appeared to uniquely represent a meiotic cell fate in honeybee. To evaluate this hypothesis, we plotted the relative expression levels of 18 genes putatively involved in meiosis II cell cycle process (GO: 0061983) as a function of pseudotime. These genes had a relatively high level of expression at the branch point, suggesting an active event of meiotic division II at this time (Fig 3B up). Among these genes, *LOC727405* showed the clearest expression peak at the branch point, resembling the dynamic pattern of its *Dmel* ortholog, *mud*, around meiosis II (Fig 3B down). The *mud* gene encodes an essential component of the meiosis II spindle apparatus in oocytes, and interestingly, its proteins are prominently present at the center of polar bodies [51]. The *mud* gene is also expressed in the *Dmel* spermatozoa although this functional role is currently unknown [52]. Thus, *LOC727405*/*mud* exhibits gene expression behavior supportive of an orthologous nature functioning in the conserved events of meiosis. Together, these results suggest that the sSPT cluster, which was identified as small spermatids for their transcriptomic similarity with late spermatocytes and their low transcriptional activity, behaved as sibling cells of normal spermatids.

To probe the transcriptional characteristics of sSPTs, we obtained the dynamic profiles of a total of 2,247 genes that were differentially expressed between the two branches (Fig 3C). Hierarchical clustering revealed four major gene groups with distinct patterns (Fig 3C and 3D). Group #1 (268 genes) showed high expression in early SPCs and top enrichment in system development (GO: 0048731), positive regulation of DNA-templated transcription (GO: 0045893), and longevity regulating pathway (KEGG: ame04213). This result is consistent with the strong transcriptional activities during this stage in both our honeybee data (Fig 1G) and previous *Drosophila* study [50]. Group #3 (794 genes) was enriched with genes in cellular component organization or biogenesis (GO: 0071840) and endomembrane system (GO: 0012505), representing the transcriptional events during the SPC-to-SPT and SPC-to-sSPT transitions. Group #4 (515 genes) showed high expression in SPTs but significantly lower expression in the sSPT branch. It was enriched with genes in ribosome (KEGG: ame03010, GO: 0005840) and oxidative phosphorylation (KEGG: ame00190, GO: 0006119). This finding suggests that in 3d honeybee adult testes some SPTs are produced through ongoing spermiogenesis, which requires high levels of mRNA translation and mitochondrial ATP synthesis. Group #2 (670 genes) with sSPT-specifically high expression was significantly enriched with genes in cilium (GO: 0005929), glycolysis/gluconeogenesis (KEGG: ame00010), and pyruvate metabolism (KEGG: ame00620). Thus, the two postmeiotic spermatid cell types exhibit distinct transcriptomic and consequent metabolic features as a function of developmental time.

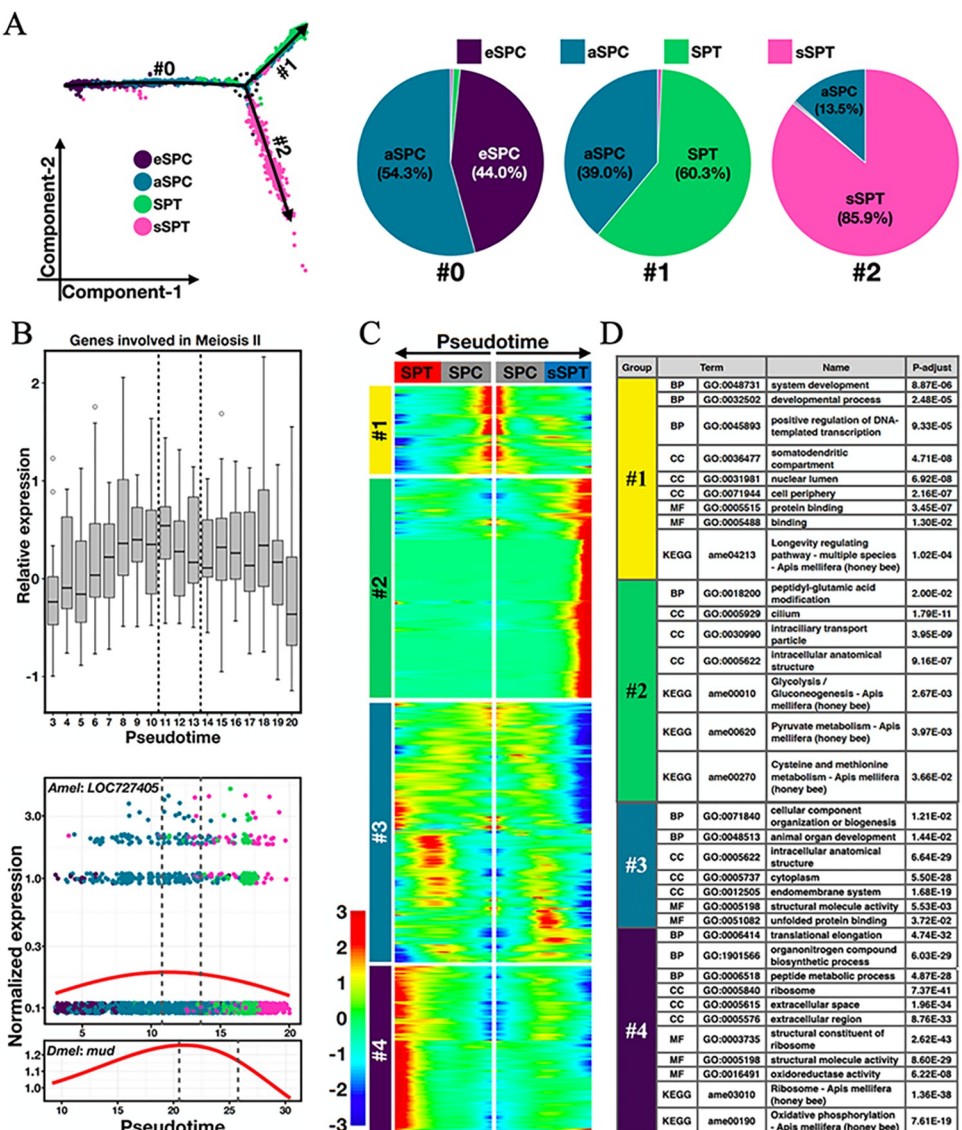

**Fig 3. Progression through meiosis is characterized by a bifurcation point.** (A) Pseudotime trajectory analysis of 4,550 meiotic cells identifies two lineages bifurcated from a branch point (dashed circle). Piecharts show the percentages of each cell type in the two lineages. (B) Scaled expression of genes in meiosis II (18 genes; upper panel) and one representative gene *LOC727405/mud* (lower panel). Cells are binned with a width of one pseudotime unit. For every gene, the *y*-axis value corresponds to *z*-scored deviation from its mean scaled expression in a bin of cells. For *Amel*, the location of the branch point (between dashed lines) was estimated from the overlapped pseudotime points among the three different branches. For *Dmel*, the dashed lines represent a transition point estimated from the overlapped pseudotime points between late spermatocytes and early spermatids in the scRNA-seq data of [50]. (C) A pseudotime-ordered heatmap from trajectory in panel A with 2,247 differentially expressed genes (adjusted *p*-value < 0.05 calculated by Monocle2) between the two bifurcated lineages. Hierarchical clustering reveals four major gene groups with distinct patterns. (D) GO and KEGG terms with top enrichment with genes in each hierarchical group of panel C.

## Hymenoptera-restricted genes are highly expressed in small spermatids

During the gene ontology analysis (see above), we observed that a number of genes with enriched expression in sSPTs (i.e., Group #2) could not be annotated by HymenopteraMine (Materials and Methods). We suspected a contribution of non-conserved genes to the

transcriptomic features of sSPTs. To test this hypothesis, we examined the orthology between *Amel* and *Dmel* for genes in Group #2. We found that 44.5% of Group #2 genes share orthology with *Dmel* genes, and the proportion is significantly lower than any other group of genes (Fisher's exact tests $p < 10^{-17}$; S5 Fig). This result suggests that the genes dictating the sSPT transcriptome do not share common ancestry with Diptera.

To better characterize the genes of different taxonomical levels that contribute to the sSPT-specific transcriptome, we categorized the *Amel* genes into four hierarchical age classes: pre-Hymenoptera (N = 8,728), post-Hymenoptera (N = 604), post-Apidae (N = 126) and Novel (N = 155), based on identifiable orthology across the animal kingdom (Fig 4A; see Materials and Methods). We examined the distribution of different gene age classes in different gene groups that show distinct meiotic cell type-specific expression patterns. Fig 4B shows that, compared to other gene groups, the sSPT-high genes (Group #2) exhibit a significantly lower fraction of pre-Hymenoptera genes and a higher fraction of Hymenoptera-restricted genes (HRGs, including the three evolutionarily younger classes). We expanded this analysis by comparing the DEG sets of the nine testicular cell types identified by our scRNA-seq, to arrive at similar result of detecting an enrichment of HRGs in sSPTs (Fig 4C). Together these results suggest that Hymenoptera-restricted genes have a relatively high contribution to the transcriptome of small spermatids.

It is known that the expression of evolutionarily young genes is overall low but exhibits sex and tissue biases [53–55]. In both whole-body and whole-testis bulk RNA-seq datasets of honeybee, we confirmed that the expression of HRGs is significantly lower than that of pre-HRGs and male-biased genes (Fig 4D–4E). However, in the single-cell cluster sSPT, the situation depended on whether the genes were highly expressed. For genes that were not highly expressed in sSPTs (i.e. not in Group #2), the transcriptional activities (measured as nUMI) of HRGs were significantly lower than those of pre-HRGs (Wilcoxon rank sum test $p = 10^{-41}$; Fig 4F right). But for sSPT-high genes (i.e. Group #2), HRGs and pre-HRGs were transcribed at a similar level (Wilcoxon rank sum test $p = 0.78$; Fig 4F left). In addition, HRGs exhibited stronger overall caste bias and tissue bias than pre-HRGs (Fig 4G and 4H comparing green to yellow), and sSPT-high HRGs displayed similar caste bias and even stronger tissue bias (Fig 4G and 4H purple). Thus, despite the generally low optimality for expression, a few Hymenoptera-restricted genes exhibit a notable sign of specific upregulation during the development of small spermatids in honeybee.

## Transcription regulatory network uncovers transcription factors that contribute to small spermatid specification

To understand the regulatory mechanisms operative in small spermatids, we constructed transcription regulatory networks (TRN) with our scRNA-seq data of 3d adult testes (Materials and Methods). Briefly, the information about transcription factors (TFs) and their DNA binding motifs were obtained from CIS-BP [56], with the testis-specific transcription start sites (TSS) identified from our full-length RNA-seq data (S3 Table). The TF-target gene pairs were scored (log-likelihood) by Cluster-buster [57] and the TRNs of sSPTs and SPTs were inferred by GENIE3 [58]. According to the inferred TRNs, the majority of active transcription factors were shared between SPTs and sSPTs (119/136 and 119/139, respectively; Fig 5A), in consistency with their transcriptome-wide correlation (Pearson's $R = 0.81$, Fig 1H). Interestingly, despite the similar numbers of total edges, there was an elevated engagement of HRGs in sSPTs relative to SPTs (Fisher's exact test $p = 0.08$; Fig 5B). Indeed, compared to pre-HRGs, HRGs are more tightly connected to the active genes in the sSPT network but not in the SPT network (Fig 5C). These results confirm that evolutionarily younger genes contribute to a

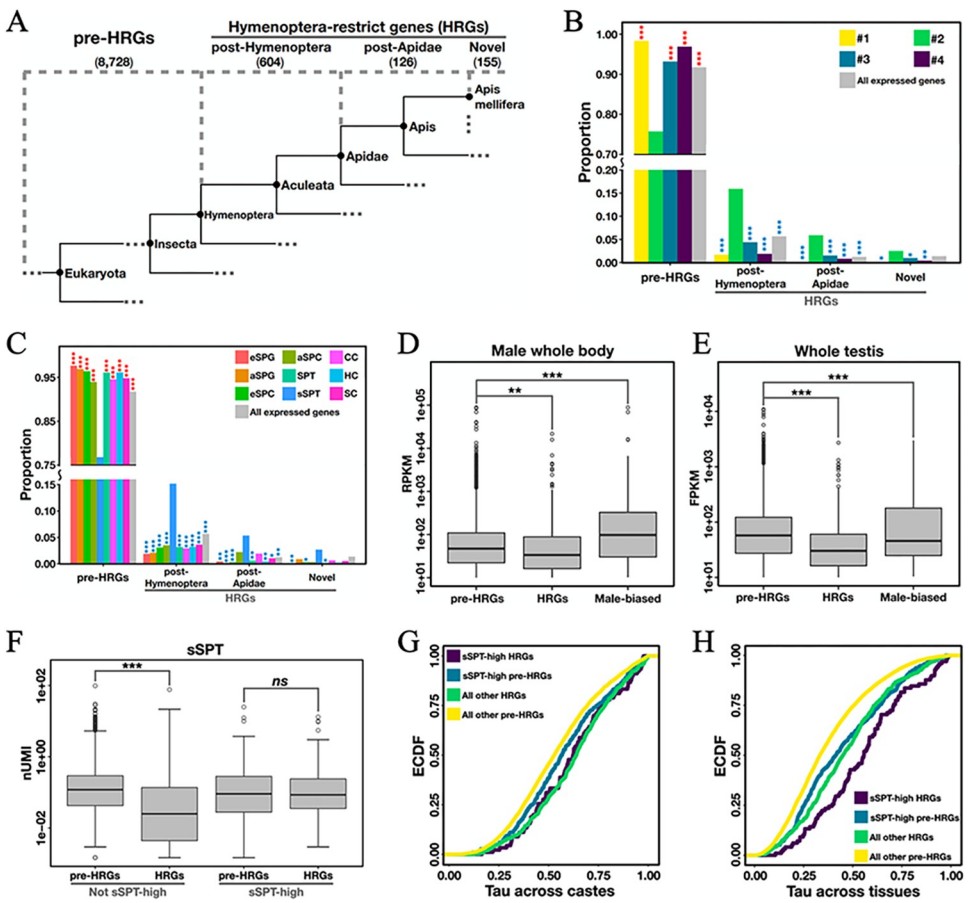

**Fig 4. Hymenoptera-restricted genes are highly expressed in small spermatids.** (A) Phylostratigraphy groups the *Amel* genes into four evolutionary age classes: pre-Hymenoptera (N = 8,728), post-Hymenoptera (N = 604), post-Apidae (N = 126) and Novel (N = 155). The three younger classes combined are referred to as Hymenoptera-restricted genes (HRGs), and the pre-Hymenoptera class is referred to as pre-HRGs. (B) Proportion of each gene age class in different gene expression groups that show distinct meiotic cell type-specific patterns (Groups #1~#4 in Fig 4D). *, **, *** denote FDR-adjusted Fisher's exact test $p < 0.05$, $< 0.01$ and $< 0.001$, respectively. (C) Same as panel B, but the gene expression groups are formed according to differentially expressed genes of the nine testicular cell types (DEGs, determined by Seurat). (D-E) Expression distribution of HRGs, pre-HRGs and male-biased genes, measured by whole-body RNA-seq (D) and whole-testis RNA-seq (E). Only genes with a relatively high expression in the datasets were used (RPKM or FPKM > 10) so that the results could be more appropriate compared with (F). Male-biased genes were defined as differentially expressed genes in drones in comparisons to their expression in queens and workers (adjusted $p < 0.05$ and $|\log_2 \text{foldchange}| > 2$). *** and ** denote Wilcoxon rank sum test $p < 0.001$ and $p < 0.01$, respectively. (F) Expression distribution of HRGs and pre-HRGs in small spermatids, measured as nUMI in testis scRNA-seq. Both HRGs and pre-HRGs are divided into two groups according to their expression in small spermatids. "sSPT-high" and "Not sSPT-high" denote genes in or out of Group #2 of Fig 4D, respectively. "ns" denotes S Wilcoxon rank sum test p > 0.05. (G-H) Empirical cumulative density function showing caste specificity (G) and tissue specificity (H) of the expression of sSPT-high HRGs (purple), all other HRGs (light green), sSPT-high pre-HRGs (dark green) and all other pre-HRGs (yellow). The tau method was based on a gene's expression difference across castes (queen, worker and drone) or across adult tissues (head, thorax, abdomen and testis). A tau of 1 indicates expression in only one caste/tissue, while a tau of 0 indicates equal expression in every caste/tissue [87].

greater degree to the transcriptional regulatory network of small spermatids than to that of spermatids.

In the sSPT network, 5 out of the 139 active TFs had target genes with relatively high HRG/ pre-HRG ratios (outside of the 95% confidence interval; Fig 5D). They were *Foxp* (encoding a Fork head-box TF, orthologous to *Dmel\FoxP*), *LOC411104* (a Homeobox TF, *Dmel\scro*), *LOC552797* (an ETS-domain TF, *Dmel\edl*), *LOC100576369* (an ETS-domain TF, *Dmel\aop*),

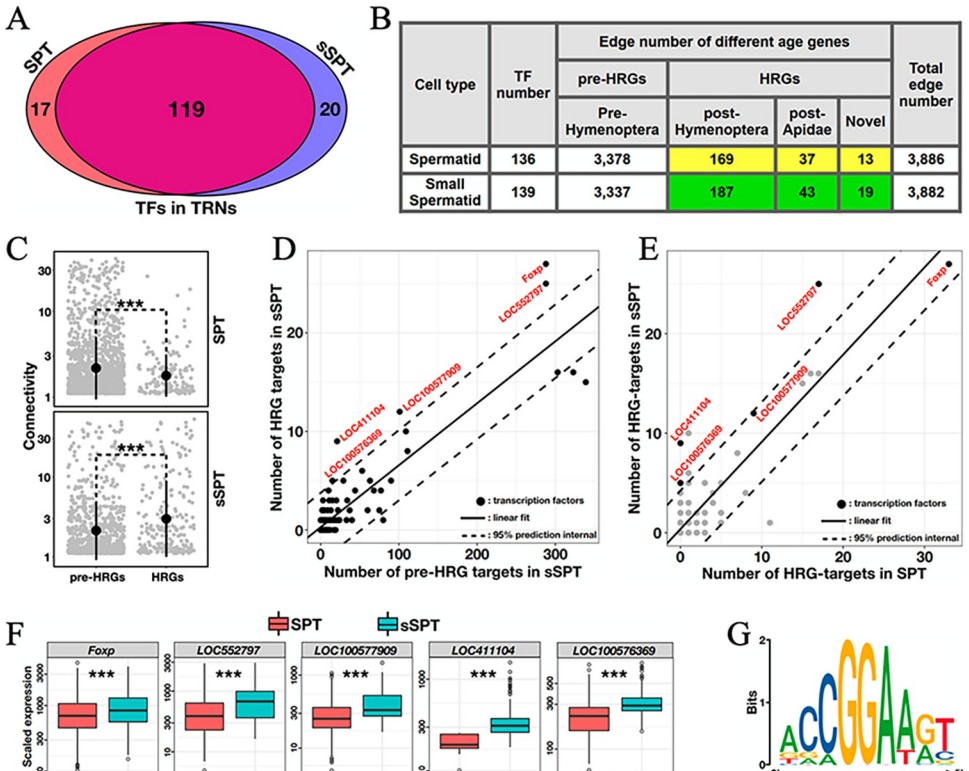

**Fig 5. Transcription regulatory network uncovers transcription factors that contribute to small spermatid specification.** (A) Intersection of active transcription factors (TF) inferred from SPTs and sSPTs. (B) Inferred numbers of regulatory edges that connect target genes of different evolutionary ages. Edges connecting Hymenoptera-restricted genes (HRG) in sSPTs are more than those in SPTs (Fisher's exact test $p = 0.08$). (C) Connectivity measured as the sum of the sixth-powered Pearson correlation between a given gene and all the other genes in the network. Student's t-tests were performed, suggesting that HRGs have a higher overall connectivity than pre-HRGs in sSPTs ($p = 10^{-5}$). (D) The numbers of HRGs and pre-HRGs that are targeted by each TF in the sSPT network. Five TFs with significantly enriched HRG targets in sSPTs are labeled in red. (E) The numbers of HRGs that are targeted by each TF in the SPT and sSPT networks. (F) Scaled expression (measured as UMI counts per million total counts in each cell, CPM) of the five TFs. *** indicates significantly (Student's t-test $p < 0.001$) higher expression in sSPTs than that in SPTs. Cells with no detection were excluded from this analysis. (G) The consensus recognition motif of the two ETS-domain TFs, LOC552797 and LOC100576369, on the target genes in the sSPT network.

and *LOC100577909* (predicted as high mobility group protein I). Among them, *LOC552797* and *LOC100576369*, each encoding an ETS-domain TF, showed both sSPT-specific enrichment of HRG targets (Fig 5E) and sSPT-specific expression (Fig 5F). An enrichment analysis also revealed a classic ETS recognition motif from the target genes in the sSPT network, C/AGGAA/T (Fig 5G). These results suggest potential roles of conserved ETS-domain factors in activating Hymenoptera-restricted genes during small spermatid specification.

## Spermatogenesis is largely absent in 14-day-old adult honeybee testes

To examine the fates of different testicular cell types, we collected testes from drones at 14 ± 1d after emergence, an age reaching sexual maturity. Consistent with a previous report [59], the size of 14d testis was much smaller than that of 3d testis (Fig 6A). Besides, while the spermatogonia and spermatocyte cysts were virtually absent in the 14d testis, there remained very few intact seminiferous tubules (Fig 6B left) and a large number of individualized spermatids were pooled together at the basal side (Fig 6B right). In fact, in 7~8d testes, the spermatogenic cysts

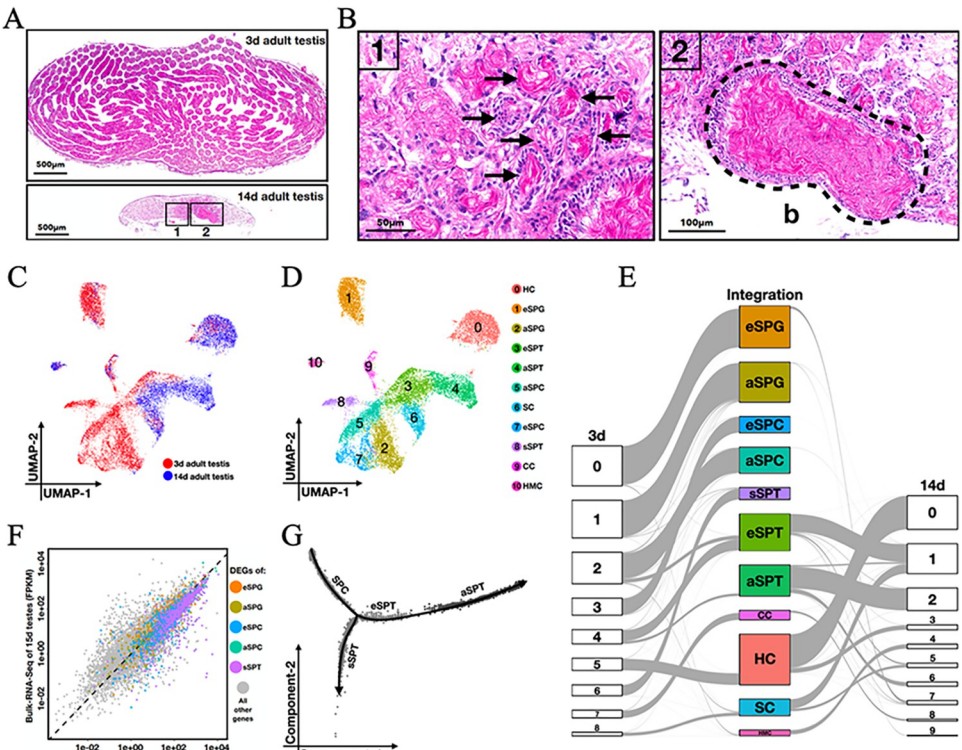

**Fig 6. Spermatogenesis is almost abolished in 14-day-old adult honeybee testes.** (A) Midsagittal sections of 3d (up) and 14d (bottom) adult testes. Hematoxylin and Eosin staining were used. (B) Zoom-in views showing degenerated seminiferous tubules (left panel) and a pool of individualized spermatids (right panel) and in the same 14d adult testis. (C) A UMAP projection of Harmony-integrated scRNA-seq datasets prepared from 3d (red) and 14d (blue) adult testes. (D) Cell type identities of the integration clusters. (E) A riverplot showing the contributions of 3d and 14d data to each integration cluster. (F) A scatterplot showing the whole-testis gene expression (measured by bulk mRNA-seq FPKM) in 3d and 14d drone adults. (G) Pseudotime trajectory analysis of all SPC, eSPT, aSPT and sSPT cells.

were still present, while the bundled spermatids massively underwent individualization (S6 Fig). These results suggest that the exact time of the complete abortion of meiosis II is likely within the window of 8d to 14d. To gain insight into the molecular events in the 14d testis, we compared bulk mRNA-seq data of 14d adult testes with that of 3d. We identified 2,119 upregulated and 1,791 downregulated genes (adjusted $p < 0.05$ and $|log_2 foldchange| > 1$ in DESeq2). While the upregulated genes were top enriched in transcription related functions such as "regulation of transcription by RNA polymerase II" (GO:0006357), the downregulated genes were enriched in ribosome (GO: 0000313, GO: 0005761) and oxidative phosphorylation (KEGG: ame00190, GO: 0006119; see S4 Table for a complete list of GO terms with enrichment). The reduction in the expression of ribosomal proteins and oxidative phosphorylation genes may be critical to the production of healthy spermatids at this sexual maturation stage.

We generated scRNA-seq data of 14d testes. Using the marker genes established in the analysis of the 3d data, we successfully identified two SPT clusters and one HC cluster, consisting of a total of 6,020 cells (S7 Fig). To delineate shared and age-specific features of cell identity between 3d and 14d testes, we integrated the two scRNA-seq datasets (Fig 6C). The joint clustering identified a total of 11 clusters, in 9 of which the cell types could be determined according to the marker genes for the 3d dataset (Fig 6D). One 14d-exclusive cluster was identified as hemocytes (HMC) with high expressions of marker genes, *LOC411597* (*Hml*) and *LOC413303* (*Pvr*) [60–62]. This other unannotated cluster, 69.8% consisting of the 14d cells, was identified

as advanced/late spermatids (aSPT) with high expressions of marker genes, *LOC724443* (*Dmel\nes*), *LOC412084* (*Dmel\f-cup*), *LOC410263* (*Dmel\Marf*), *LOC409838* (*Dmel\Gld2*), *LOC411115* (*Dmel\Bruce*), *LOC551259* (*Dmel\sls*) and *LOC551356* (*Dmel\Msp300*) [63–69]. A riverplot shows the assignments of each integration cluster to the two datasets, revealing a lack of early spermatogenic processes and an accumulation of late spermatid cells in 14d testes (Fig 6E). This finding was further verified by the bulk mRNA-seq detection of a general decrease in the maker genes for early spermatogenic cell types, including eSPGs, aSPGs, eSPCs and aSPCs (Fig 6F). Interestingly, sSPTs were also completely absent from the 14d dataset. Pseudotime analysis of all SPC, eSPT, aSPT and sSPT cells failed to recover a trajectory between sSPTs and aSPTs (Fig 6G), suggesting that sSPTs likely have been eliminated from the testes before 14d of age.

## Mutation load is decreased in spermatids of 14-day-old adult honeybee testes

Since germline mutations can impact both reproductive outcome and evolutionary innovation, we inquired about the changes in the mutation load across different spermatogenic cell types and between 3d and 14d testes. By contrasting the variants called from each germline cell type with those called from somatic cell types in our scRNA-seq datasets (and filtering according to several other criteria; Materials and Methods), we identified a total of 4,031 high-confidence substitutions that are putatively *de novo*. We found that while eSPCs carried the greatest number of substitutions on average (0.84 substitutions per cell), the rate declined to below 0.2 in later meiotic cells (Fig 7A). We also approximated the per-base rates for every germline cell type as previously described [50]. We detected a decline from eSPCs to aSPTs (Fig 7B), a generally similar trend as in *Dmel* [50]. By contrast, sSPTs had a higher per-base rate than either eSPTs (by 2.9 fold) or aSPTs (3.7 fold).

To evaluate the contribution of adult age to the mutation load, we separated 3d data from 14d data in eSPTs, and aSPTs. We found that, for both eSPTs and aSPTs, the per-base substitution frequencies were >7 fold lower in 14d cells than in 3d cells (Fig 7C). Interestingly, we observed an expression bias of DNA repair genes towards 14d cells and early spermatids, exhibiting a reverse trend relative to the mutation rates in these cell groups (Fig 7D). Thus, elimination of small spermatids that carry high mutation load and elevation of the DNA repair activity may contribute to safeguarding sperm quality of honeybee testes during the sexual maturation stage.

## Discussion

Single-cell RNA sequencing represents a powerful approach toward capturing rare cell populations and better understanding of gene activities during developmental processes. Recent studies have generated single-cell transcriptomic maps for spermatogenic and testicular somatic cells in humans and several model organisms [49,70–76]. Comparative analyses on testis scRNA-seq datasets have elucidated that spermatogenesis is a highly-regulated process under the control of regulatory pathways that are evolutionarily conserved [74,77]. Indeed, even phylogenetically distant species can share genetic mechanisms that underlie different steps of germ cell production, including mitosis, meiosis and maturation [78–81]. For example, 306 out of 379 *Dmel* genes whose mutation gives rise to a defective phenotype of male reproduction possess at least one mouse (*Mus musculus*, *Mmus*) ortholog, with a conservation percentage (80.7%) almost twice as high as the rate (41.7%) for the whole *Dmel* genome [81]. Meanwhile, 277 out of these *Dmel* genes (73.1%) possess at least one *Amel* ortholog [78]. In this study, we generated scRNA-seq data from honeybee adult testis samples and have

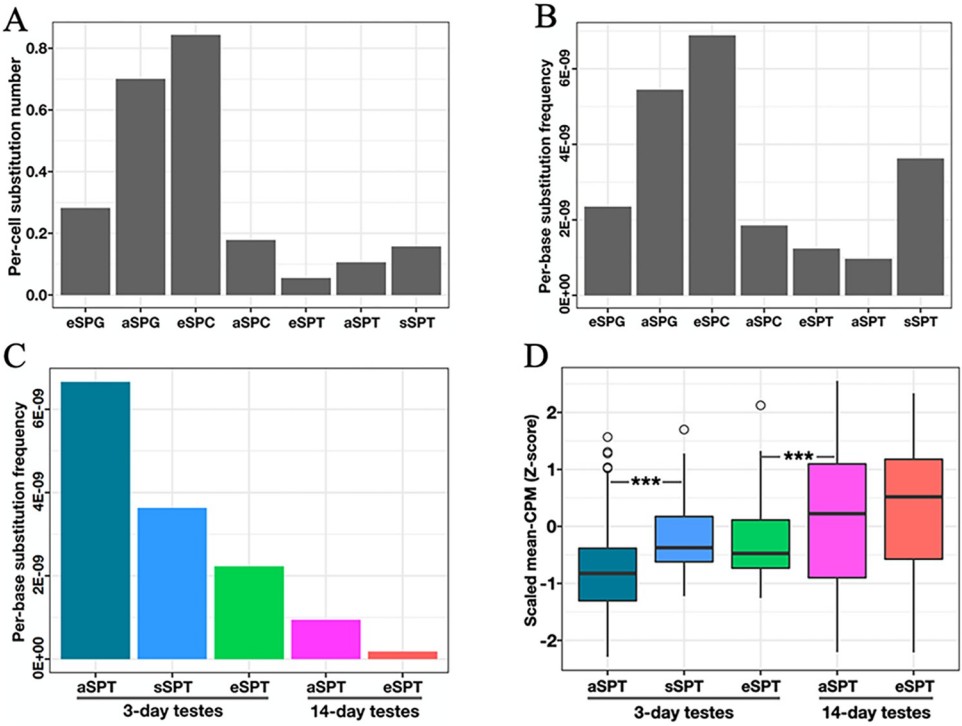

**Fig 7. Late spermatids in 14-day-old adult honeybee testes carry more substitutions.** (A) A total of 4,031 high-quality substitution variants were identified from our scRNA-seq datasets (Materials and Methods). Shown are the numbers of substitution events per cell for every germline cell type. (B) Per-base substitution frequency for each cell type was estimated by dividing the per-cell rate by the number of bases covered with at least 10 reads in that cell type. (C) Per-base substitution frequencies in 3d eSPTs, 3d aSPTs, 3d sSPTs, 14d eSPTs and 14d aSPTs. Data were shown in a descending order. (D) Scaled expression of DNA repair genes (GO: 0006281). Student's t-tests were performed.

successfully identified the major cell types of spermatogenesis with marker genes that share 1-to-1 orthology between *Amel* and *Dmel*. Thus, our work paves the way for future investigations on the underlying mechanisms of male reproduction in honey and other Hymenopteran species.

Our results show that in newly-emerged (3d) honeybee drones, spermatogenesis remains ongoing at a single-cell resolution. In fact, a honeybee adult testis consists of 150~200 seminiferous tubules, among which the developmental stages vary significantly in a manner that is dependent on their locations within the testis. There is also notable asynchrony in cell cycle progression between different germ cell cysts within the same seminiferous tubule [3,13]. These results argue against the idea that all germ cell cysts from drones at this adult age have already concluded spermatogenesis. Our measurement of spermatozoon numbers in adult drones of accurately-determined ages verifies new sperm production in the first two days after emergence (Fig 1C). Intriguingly, a group of cells, which represent the earliest spermatogonial stage in the 3d testes, exhibits an enrichment of genes involved in germline stem cell maintenance. These molecular markers will benefit future efforts in defining the spatiotemporal distribution of germline stem cells during the testis development of honeybee.

Our data captures a unique type of cells, small spermatids, which exhibit properties consistent with an origin from an uneven meiosis II in the 3d testes. Small spermatids, together with early-stage cell types including spermatogonia and spermatocytes, are mostly absent in the fully-matured (14d) drones. These results are consistent with a previous hypothesis that small spermatids simply died out [6,10,15,82]. In addition, small spermatids have a mutation load

higher than that in other spermatids (Fig 7A and 7B), supportive of a unique system in the honeybee drones to produce a desired number of high-quality mature gametes.

While this work has generated useful data and insights, it is currently limited by the accessibility of healthy drones for accurately determining the age and collecting samples. The marker genes identified by our analysis are expected to exert biological functions conserved between the honeybee and *Drosophila* to support the conserved events of spermatogenesis. While this notion remains to be validated experimentally in future studies, our informatics analysis, as an alternative—though imperfect—to experimental validation, has revealed that the bee-specific small spermatids are characterized by a specifically enriched expression of Hymenoptera-restricted genes. On one hand, this result helps rule out the possibility that the identified small spermatid cells were cytoplasmic fragments or blebs derived from other types of cells. On the other hand, while our results suggest that Hymenoptera-restricted genes might have their functions at the time when small spermatids arose as a new cell type during evolution, the question of why the bees uniquely produce small spermatids remains to be understood. Furthermore, it will be insightful to determine the drone age at which meiosis II becomes completely aborted and when small spermatids get eliminated. Such determination will require future studies make use of single-cell analysis of the honeybee spermatogenesis across more developmental stages, including the intermediate drone ages between emergence and maturation. Despite the noted limitations, our study represents the first single-cell transcriptomic analysis of spermatogenesis in the honeybee, paving the way for future mechanistic studies.

## Materials and methods

### Collection and dissection of honeybee drone adults

The honeybee (*Apis mellifera ligustica*) colonies used in the study were maintained by Institute of Apicultural Research (IAR) Chinese Academy of Agricultural Sciences (IAR) in Beijing 100093, China. During May 2019, apparently healthy drones within 24 hours after emergence (day 0) were selected according to body size and absence of pathological signs such as trembling, deformed wing and swollen abdomen as described in [13]. These drones were marked on the thorax, then reared in the original hives, and finally collected on different days. The collected drones were dissected as described in [83]. Then the total numbers of spermatozoa were respectively counted from freshly dissected testes and seminal vesicles using Countstar Rigel S2 Fluorescence cell analyzer (Alit Biotech-Shanghai).

### Testis sectioning and H&E staining

Freshly dissected testes of drone adults were fixed in 4% paraformaldehyde (pH 7.4), washed and stored in 0.1 M PBS (pH 7.4) at 4°C. Sagittal sections were obtained at 4-μm thickness, deparaffinized, rehydrated and stained with hematoxylin and eosin according to the manufacturer's instruction (ZKWB-Bio).

### Preparation and sequencing of testis scRNA-seq libraries

We generated scRNA-seq datasets for 3d and 14d adult honeybee testes, respectively. Freshly dissected testes from 6 drone adults were incubated in lysis buffer with 1.6 ml TrypLE (ThermoFisher) and 0.4 ml collagenase Type I (GIBCO) at 28°C with gentle vortex for 30 min and 37°C for another 30 min. The samples were repeatedly filtered through a 30 mm nylon mesh with addition of cold Hanks' Balanced Salt Solution (HBSS), and then centrifuged for 5 min at 300 RCF (Eppendorf 5804R). The cells were washed in cold HBSS and pelleted by

centrifugation for three rounds. The resulting cell preparation was resuspended in 150 μL cold HBSS before further processing. For cell counting and viability estimation, 9 μL of the cell suspension was mixed with 1 μL 0.4% Trypan Blue for 3 min, loaded to a hemocytometer and imaged by an upright light microscope (Countstar Rigel S2). The results confirmed that our approach yielded a high number of single cells with an average viability of 95% (S1A Fig). According to 10X Genomics protocol, we diluted the cell suspension to a final concentration of 1,200 cells/μL and used Chromium Single Cell 3' Reagent Kits v3.1 for library preparation, followed by sequencing with Illumina NovaSeq 6000 paired-end chemistry.

### Processing of scRNA-seq data

Reads were processed using cellranger v3.1.0 with all default parameters and the RefSeq assembly of GCF_003254395.2_Amel_HAv3.1 as the reference genome. A total of 20,436 and 11,097 cells were retrieved from the 3d and 14d datasets, respectively. The filtered matrices were imported to Seurat v4.0. From each dataset, we removed genes expressed in fewer than three cells and cells expressing fewer than 200 genes. We used Seurat to normalize and scale the gene expression data and run PCA with top $a$ (800~1,100) highly variable features (HVFs). We then used Seurat to perform unsupervised clustering with top $b$ (7~13) PCs and resolution $c$ (0.1~0.6). Of all the tested parameter sets {$a$, $b$, $c$}, we observed a cluster (cluster-null in S1B–S1D Fig) in the center of the UMAP pattern, which occupied a high fraction in the entire dataset and contained a high percentage of relatively low-complexity cells (gene number < 2,000 and UMI < 5,000). We suspected that the cluster-null cells were resulted from debris and small blebs of cytoplasm (e.g. interzonal bodies), and excluded 8,779 and 3,700 cluster-null cells from the two datasets, respectively, for further analysis. In addition, we used DoubletFinder v2.0.3 with default parameters to identify doublet signals (874 and 555 cells in the two datasets, respectively; S1E Fig). After removal of the doublets, we recovered a total of 10,783 and 6,842 high-quality cells from the two datasets, respectively. For these high-quality cells, we again used Seurat to run normalization, scaling and unsupervised clustering. Of the tested parameter sets {$a$, $b$, $c$}, all produced a similar UMAP pattern, but {825, 8, 0.17} generated the best separation between different clusters and the least ambiguity of cell types at the following step. Under the same setting, we obtained a total of 9 and 10 cell clusters from the 3d and 14d datasets, respectively.

### Inference of the major cell types in the 3d dataset

To infer the cell types in our honeybee datasets, we relied on developmental biology of *Dmel* and gene orthology between *Dmel* and *Amel*. We curated a collection of 248 genes that were specifically expressed in certain cell types of the *Dmel* testis (S1 Table). We combined OrthoDB and OrthoFinder results to determine that 135 *Amel* genes are one-to-one orthologs to the 248 *Dmel* genes. We verified the expression of these genes in our datasets and used two criteria to determine whether their expression was enriched in specific single-cell clusters. We required a candidate marker gene to be expressed in more than 10% cells of the cell cluster it marks. We also required the marked cluster to express the candidate marker gene at a level higher than 50% of the maximum level among all clusters. Thus, we obtained a total of 109 *Amel* genes with *Dmel* cell type information and *Amel* cell cluster enrichment (Fig 1E and S1 Table). To label a given *Amel* cell cluster with a specific *Dmel* cell type, we counted the number of all candidate marker genes that were enriched in this cluster and supported this cell type. According to these numbers, we identified 8 of the 9 clusters in the 3d dataset as early spermatogonia (eSPG), advanced/late spermatogonia (aSPG), early spermatocytes (eSPC), advanced/late spermatocytes (aSPC), spermatids (SPT), hub cells (HC), cyst cells (CC), and sheath muscle cells

(SC), respectively. The remaining one cluster could not be properly defined with the same procedure. We suggest that this cluster might resemble the small spermatid cells (sSPT), which have been reported in previous anatomic studies [11], under the considerations about their transcriptomic similarity with aSPCs and their low transcriptional activity (see more details in the main text). After all the single-cell clusters were typed, we iteratively removed candidate marker genes that exhibited inconsistency or high ambiguity. For a good marker gene, if it marked only one *Amel* cell type, their correspondence must be supported by the *Dmel* knowledge in S1 Table. If it marked two or more *Amel* cell types, the correspondences not supported by the *Dmel* knowledge must not exceed two. We obtained a final set of 64 *Amel* marker genes that could sufficiently identify a total of 9 cell types in the 3d dataset.

## Integration and contrast of the 3d and 14d datasets

We integrated the Seurat objects of the two scRNA-seq datasets using Harmony v1.0 with top 25 PCs and lambda = 0.3. Then we used top 10 Harmony-calculated PCs to perform Seurat clustering with resolution = 0.5, and obtained a total of 11 integration clusters. We calculated the contribution of each cell type of the 3d adult testis in each integration cluster. We found that 8 out of the 11 integration clusters could be properly defined without ambiguity according to the 3d cell types, including eSPG, aSPG, eSPC, aSPC, sSPT, HC, CC and SC. Two integration clusters were contributed by SPT cells of the 3d dataset. While the integration cluster-3 contained 660 SPT cells, 237 SPC cells and 47 other 3d testicular cells, the integration cluster-4 contained 166 SPT cells and 6 other 3d testicular cells. Considering that the integration cluster-3 is located in between aSPC (integration cluster-5) and the integration cluster-4 in the UMAP pattern (Fig 6D), we denoated the integration cluster-3 and cluster-4 as early spermatids (eSPT) and late spermatids (aSPT). In fact, 69.8% of the aSPT cluster consisted of 14d cells, expressing a high level of several marker genes for individualized spermatids such as *LOC724443* (*nes*), *LOC412084* (*f-cup*), *LOC410263* (*Marf*), *LOC409838* (*Gld2*), *LOC411115* (*Bruce*), *LOC551259* (*sls*) and *LOC551356* (*Msp300*). Another integration cluster, also predominatly (75.7%) contributed by 14d cells, was identified as hemocytes (HMC) according to the marker genes *LOC411597* (*Dmel\Hml*) and *LOC413303* (*Dmel\Pvr*).

## Classification of gene evolutionary ages

By regrouping the phylostratigraphic categories as previously determined [84], We focused on four hierarchical levels: "pre-Hymenoptera" (8,728 *Amel* genes displaying orthology with non-hymenopteran animals), "post-Hymenoptera" (604 *Amel* genes found in non-Apidae hymenopterans but not in any other species), "post-Apidae" (126 *Amel* genes found in non-*Amel* bees but not in any other species), and "Novel" (155 genes found only in *Amel*).

## Bulk mRNA-seq datasets and their processing

We generated whole-testis bulk mRNA-seq libraries of 3d honeybee drones, 14d honeybee drones and 14d bumblebee (*Bter*) drones, respectively (two independent replicates each). The sequencing was performed with PE150 chemistry on Illumina NovaSeq 6000. The whole-testis mRNA-seq datasets of *Dmel*, *Sinv* and *Nvit* were obtained from www.ncbi.nlm.nih.gov/geo (SRR17400003...SRR17400005, SRR6926147...SRR6926148, and SRR12922171...SRR12922173). Nine whole-body mRNA-seq datasets of honeybee queen, worker and drone adults were obtained from www.ncbi.nlm.nih.gov/geo (SRR7143860...SRR7143868). Twelve mRNA-seq datasets of honeybee drone head, thorax and abdomen parts were obtained from ngdc.cncb.ac.cn/gsa (CRR106390...CRR106401). All bulk mRNA-seq reads were mapped to the reference genomes using hisat2 v2.2.1, and summarized

using htseq-count v1.99.2. See S6 Table for the versions of all genome assemblies and gene annotation files used in this study.

## Differentially expressed gene and functional enrichment analyses

For scRNA-seq data, differentially expressed genes were called using FindAllMarkers of Seurat. For bulk mRNA-seq data, DESeq2 was used. Gene ontology enrichment analysis was performed using the tool provided by hymenopteramine.rnet.missouri.edu/hymenopteramine, and the adjusted *p*-values were obtained using hypergeometric tests with Holm-Bonferroni correction.

## Preparation and processing of full-length RNA-seq data

Full-length RNA-seq libraries of honeybee adult testes (within 14 days after emergence) were constructed and sequenced according to the PacBio Iso-Seq2 pipeline. The raw data were processed using SMRT Link v9.0.0.92188. The classified full-length Circular Consensus Sequence (CCS) reads were mapped to GCF_003254395.2_Amel_HAv3.1 using Minimap2 v2.17 with "-ax splice", and the alignments were assembled using StringTie v2.1.4 with "-c 1.0". The StringTie assembly was analyzed using Gffcompare v0.11.7.

## Identification of honeybee adult testis-specific transcription start sites

We used the Gffcompare annotated transcriptome to build a STAR reference and mapped the bulk mRNA-seq reads of 3d and 14d adult honeybee testes to it. We required honeybee adult testis-expressed transcripts either 1) reported by Gffcompare and supported by bulk mRNA-seq FPKM > 1 or 2) annotated in the reference genome and supported by bulk mRNA-seq FPKM > 10. We merged their transcription start sites (TSS) as shown in S3 Table.

## Construction of transcriptional regulatory networks

The annotations of transcription factors (TFs) in *Amel* were downloaded from CIS-BP [56]. The gene identifiers were converted in accordance with GCF_003254395.2_Amel_HAv3.1 according to BeeBase datasets [85]. We obtained a total of 212 candidate TFs and the information of their DNA binding motifs characterized as position weight matrix (PWM). Then we used Cluster-buster [57] to find and score clusters of these motifs within the genomic region of -3~+1 kb surrounding each adult testis-specific TSS (S3 Table). For each pair of candidate TF and its putative target gene, the highest score among all found motif clusters was considered as the score of their regulatory relationship. We used GENIE3 v.1.16.0 [58] to infer transcriptional regulatory networks of the SPT cells and the sSPT cells, respectively. Raw UMI counts (nUMI) were used as the input expression matrix. To build the final regulons, we required all edges with a minimum weight of 0.02, and we kept only 5% of the target genes for each TF with top Cluster-buster scores. We used "connectivity" to quantify how well a given gene was connected within the gene network as described in [84]. It was calculated as the sum of the sixth-powered coefficients of Pearson's correlation between a gene and all the other genes.

## Variant calling from scRNA-seq data

We ran mpileup (bcftools v1.8) with a minimum quality of 30 to find nucleotide polymorphisms from demultiplexed reads of our scRNA-seq data. We discarded the calls that meet the following criteria: 1) indels, 2) variants with a sequencing depth < 10 or >50, 3) variants detected in five DNA-seq libraries of drone adult somatic tissues from the same apiary (one generated in the current study and four from [84], and 4) potential A-to-I editing sites, which

were identified as previously described [86]. Then, we used samjdk to split the results according to different cell types, and removed variants that could be found in somatic cells (HC, CC, SC and HMC). Considering proximal substitutions in the same cells as single events, we subtracted variants present within 10bp of each other in the same cells, and finally obtained a total of 4845 polymorphisms. To calculate per-base substitution rate for a given set of single cells as previously described [50], we divided the total number of substitutions in these cells by the number of cells and the number of bases covered by at least 10 reads from these cells (S5 Table).

## Supporting information

**S1 Fig. Filtering low-quality cells from scRNA-seq datasets of honeybee adult testes.** (A) A representative bright-field image of single-cell preps from the 3d adult testes. The preps were stained with acridine orange and propidium iodide. The images were captured and analyzed by Countstar Rigel S2. (B-C) UMAP projections of 20,436 and 11,097 cells from 3d (A) and 14d (B) adult testes, respectively. Seurat parameters: HVF = 1,050, PC = 13, and resolution = 0.5 (see Materials and Methods). Note that the sSPT cluster of cells (dashed circle) were clearly separated from cluster-null (red) in the 3d dataset. (D) A scatter plot of the proportion of low-complexity cells (gene number < 2,000 and UMI < 5,000) in each cluster against the share of this cluster in all cells. Green: 3d; black: 14d. Note that the sSPT cluster (arrow) was distant from clusters-null (solid circle). (E) Identified doublets from 3d and 14d dataset, respectively. (F) The average read qualities in the sSPT cell type, all the other *Amel* cell types and all the *Dmel* cell types identified from the data of [50]. (G) The single-cell qualities measured as the fraction of mitochondria-encoded transcripts.
(TIF)

**S2 Fig. Expression and evolution signatures of the spermatogenic cell type-specific marker genes.** (A) Expression phylogeny of the 53 spermatogenic cell type-specific marker genes based on the bulk RNA-seq data from adult testes of five species: *Drosophila melanogaster* (*Dmel*), *Apis mellifera* (*Amel*), *Bombus terrestris* (*Bter*), *Solenopsis invicta* (*Sinv*) and *Nasonia vitripennis* (*Nvit*). Bootstrap analyses (a total of 4,974 one-to-one orthologous genes were randomly sampled with replacement for 1,000 times) suggest a high reliability of the branching pattern (proportions of replicate trees > 0.9). (B) The *Amel* marker genes and their counterparts in the other four species all show an overall higher level of testicular expression. *** denotes a *p*-value < 0.001 of Wilcoxon rank sum test comparing the marker genes with all the other genes with FPKM > 1 or RPKM > 1 in the corresponding testis bulk RNA-seq datasets (Materials and Methods). (C) The distributions of $d_S$ and $d_N$ for the spermatogenic cell type-specific marker genes shared by the genera *Apis* (*Apis mellifera*, *Apis cerana*, *Apis dorsata* and *Apis laboriosa* were used in the calculation; red) and *Drosophila* (*Drosophila melanogaster*, *Drosophila simulans*, *Drosophila yakuba*, *Drosophila pseudoobscura* and *Drosophila grimshawi*; green). (D) The distributions of $d_N/d_S$ for the spermatogenic cell type-specific marker genes shared by *Apis* and *Drosophila*. Wilcoxon rank sum test *p*-value = 0.8.
(TIF)

**S3 Fig. Rates of the daily changes in testicular and seminal vesicular spermatozoon numbers in honeybee adult drones.** The raw data (N = 20 independent samples on each day) were shown in Fig 1C. Bootstrapping was performed by resampling the data for 20 times for each day. Then the change rate on each day was estimated as the slope of linear fit with two preceding data points, the current data point and the following two data points in each bootstrapping vector. Student's t-tests were used to statistically compare the change rates between seminal

vesicle measurements and testis measurements.
(TIF)

**S4 Fig. Histological sections of 3-day-old honeybee adult testes.** Hematoxylin & Eosin staining were used. Dashed circle: spermatogenic cyst; arrowhead: spermatid bundle.
(TIF)

**S5 Fig. The proportion of genes orthologous to *Dmel* genes for the four gene groups in Fig 3C and all expressed genes.** *** denotes Fisher's exact test *p*-value < 0.001.
(TIF)

**S6 Fig. Histological sections of 7-day-old and 8-day-old honeybee adult testes.** Dashed circle: spermatogenic cyst; arrowhead: spermatid bundle; double-arrow: spermatid bundle undergoing individualization; dashed rectangle: individualized spermatids.
(TIF)

**S7 Fig. Cell type annotation of unsupervised clusters in the 14d data.** (A) One cluster of hub cells and two clusters of spermatids were identified with the 64 marker genes established in the analysis of 3d data (Fig 1F). (B) The marker genes of the three cell types identified from the 14d-only data.
(TIF)

**S1 Table. Determination of honeybee testicular cell type-specific marker genes.**
(XLSX)

**S2 Table. GO and KEGG terms enriched with spermatogonially expressed genes from each hierarchical group of Fig 2C.**
(XLSX)

**S3 Table. Honeybee adult testis-specific transcription start sites.**
(XLSX)

**S4 Table. GO and KEGG terms enriched with differentially expressed genes in comparison between 3d and 14d testis bulk RNA-seq.**
(XLSX)

**S5 Table. Cell type-specific variant calling from 3d and 14d testis scRNA-seq data.**
(XLSX)

**S6 Table. A list of genome annotation files used in this study.**
(XLSX)

## Acknowledgments

We acknowledge support of Zhejiang University (ZJU) School of Medicine affiliated Women's Hospital and Children's Hospital.

## Author Contributions

**Conceptualization:** Jiaxing Huang, Jun Ma, Feng He.

**Data curation:** Zhiyong Yin, Yingdi Xue, Xianghui Yu.

**Formal analysis:** Zhiyong Yin, Feng He.

**Funding acquisition:** Jiaxing Huang, Jun Ma, Feng He.

**Investigation:** Zhiyong Yin, Guiling Ding, Jie Dong.

**Project administration:** Feng He.

**Resources:** Jiaxing Huang.

**Writing – original draft:** Zhiyong Yin, Jun Ma, Feng He.

**Writing – review & editing:** Zhiyong Yin, Jun Ma, Feng He.

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
