## [Decision Letter · Decision Letter 0]

29 Aug 2023

Dear Dr He,

Thank you very much for submitting your Research Article entitled 'A postmeiotically bifurcated roadmap of honeybee spermatogenesis marked by phylogenetically restricted genes' to PLOS Genetics.

The manuscript was fully evaluated at the editorial level and by independent peer reviewers. The reviewers appreciated the attention to an important problem, but raised some substantial concerns about the current manuscript. Based on the reviews, we will not be able to accept this version of the manuscript, but we would be willing to review a much-revised version. We cannot, of course, promise publication at that time.

If you decide to revise the manuscript for further consideration at PLOS Genetics, please aim to resubmit within the next 60 days, unless it will take extra time to address the concerns of the reviewers, in which case we would appreciate an expected resubmission date by email to plosgenetics@plos.org.

We are sorry that we cannot be more positive about your manuscript at this stage. Please do not hesitate to contact us if you have any concerns or questions.

Yours sincerely,

Takaaki Daimon

Academic Editor

PLOS Genetics

Gregory P. Copenhaver

Editor-in-Chief

PLOS Genetics

Reviewer's Responses to Questions

**Comments to the Authors:**

Reviewer #1: In this manuscript, Yi and colleagues analyze spermatogenesis of the honeybee with single-cell sequencing. Honeybees are a fascinating and understudied species, with an unusual meiotic program: haploid males, produced through asexual reproduction, fail to undergo a reduction division in meiosis to retain a haploid state in their gametes. Some data, as cited by the authors, suggest that honeybee male meiosis II is asymmetric, as reminiscent of mammalian female meiosis: each meiotic II division gives rise to one spermatid and one small haploid cell. To study this interesting phenomenon, the authors perform single-cell RNA sequencing on honeybee testes.

Because the honeybee is so understudied relative to its potential, this dataset is of great interest to germ cell biologists generally. The dataset appears to generally be of high quality, and should absolutely be published (and I think that PLOS Genetics is an undeniably appropriate journal for such a dataset to be published in). That being said, I have some concerns about the validity of the cell type annotation, which is necessary to build the subsequent conclusions of the paper.

First, the authors use 1-to-1 homology with Drosophila genes to annotate their clusters. It is unclear where the Dmel cell type analysis comes from, since the reference column in the supplemental table doesn’t appear to have a corresponding reference list (or perhaps I simply couldn’t find it). Regardless, as identified by the authors, honeybee spermatogenesis occurs very differently from Dmel, both in regards to meiotic behavior and with a lack of a single, maintained stem cell population that allows for homeostatic spermatogenic turnover throughout the fly’s lifetime. Therefore, without validation of these markers in the honeybee (through RNA in situ hybridization or other methods), it’s hard to know if the assumptions that Dmel factors have the same cell type expression is appropriate. Likewise, there’s also some mixed information about these markers: zfh1, for example, is listed as a GSC marker in the text, but as a cyst cell marker in the supplementary table.

Second, the most important conclusion of the paper comes from annotation of a cluster that lacks expression of Dmel orthologs, and is also generally UMI low. This cluster is annotated as “small spermatids,” or sSPTs. Notably, the authors themselves remove cells in the dataset with inconsistency or high ambiguity, relative to the Drosophila data, as well as removing low-complexity cells. It is easy to believe that the sSPT-annotated cells are simply low-quality or poorly sequenced cells, accounting for their low map rate and apparent high mutational rate. Alternatively, since the authors performed single-cell sequencing, rather than single-nuclei sequencing, it is very possible that this cell cluster arose from fragmenting spermatid heads from spermatid tails. In Drosophila, these cell types are very fragile, and cell suspensions can often contain fragments of spermatid tails that can be captured by droplet-based sequencing and mistaken for cells, since they do contain significant amounts of RNA. This could result in what appear to be two spermatid populations, when, in fact, it’s simply two parts of the spermatid sequenced separately. Again, without validation, it’s hard to know if this cell type is real or artifactual. Since many of the main conclusions of the paper depend on this annotation, it’s hard to asses them.

Overall, I recommend this paper for publication with significant revisions. The data is worthy and should be published, but the conclusions are overstated with the level of analysis done. Either the authors should validate their cell type annotations in vivo using their newly identified honeybee marker transcripts, or they should clearly lay out the limitations to the analysis given the potential issues with annotation.

Reviewer #2: This study utilized single-cell RNA sequencing to examine spermatogenesis in honeybee drones at two developmental timepoints, 3 days and 14 days after emergence. The authors identified distinct spermatogenic cell populations in 3-day-old drones, including a small spermatid population potentially arising from asymmetric meiotic division. They found active spermatogenesis in 3-day drones but not in 14-day drones, indicating spermatogenesis is largely complete by 14 days of age.

The experiments appear technically sound, but there are some concerns about the biological interpretations that require further clarification. The authors relied heavily on known markers from Drosophila to identify cell types. However, meiosis I and testis morphology differ substantially between Drosophila and Apis. To strengthen the conclusions, the authors should perform comparative analyses using genetic data from other hymenopteran species. This may allow better identification of honeybee-specific cell types based on evolutionary signatures rather than assuming orthology with Drosophila.

The small spermatid population is an intriguing finding. To definitively demonstrate their origin and fate, functional experiments tracking these cells over development would significantly enhance this study. Their transient presence and high mutation load suggest they may contribute to sperm quality control, but it is unclear why honeybees uniquely produce small spermatids. Further studies of their functional roles across more timepoints is needed.

In summary, this study provides valuable single-cell transcriptomic resources for honeybee spermatogenesis. However, some conclusions require more functional validation and evolutionary comparisons between hymenopteran species. Addressing these concerns would significantly strengthen this contribution.

Additional specific issues were raised:

Figure 1: The authors chose 3d and 14d drones to prepare scRNA-seq libraries from freshly dissected testes. Then identified different cell types in the testicular tissue of the 3d drones, and suggesting a continuous process of spermatogenesis presented in the newly emerged honeybee testes. The article points out that 3d of age represents a transition stage when a massive number of spermatozoa are transferred from testes to seminal vesicles. However, from the results of Figure 1B, it seems that a 4-day drone is more appropriate. Please explain it.

Figure 2: The identification of putative germline stem cell markers in early spermatogonia is an interesting finding. Further functional validation could strengthen the conclusion that these cells have differentiation potential.

Figure 3: The authors note that pseudotime trajectories for meiotic cells typically show a continuous developmental progression in mouse and fly studies. However, they identified a unique bifurcation in honeybees, with a branch giving rise to aSPC-sSPT cells. This aSPC-sSPT branch appears to represent a distinct meiotic cell fate in honeybees. Initial identification of these cell populations relied on known marker genes from Drosophila due to conserved aspects of spermatogenesis. However, the emergence of small spermatids from asymmetric meiotic division seems to be a more lineage-specific feature in bees. The authors could perhaps comment in more detail on both the similarities to Drosophila meiosis based on orthologous markers, as well as any bee-specific characteristics of the transcriptomic profiles and regulatory relationships uncovered in the aSPC-sSPT branch. This may provide better evolutionary context for interpreting this key result.

Figure 4: The authors showed that transcriptional activities of HRGs and pre-HRGs were comparable in spermatids, which differs from their expression patterns in whole-body and whole-testis samples. It would be helpful for the authors to also examine HRG/pre-HRG expression differences specifically among the highly expressed genes in the whole-body and whole-testis data. Adding this analysis would allow for a more equivalent comparison to the spermatid expression data, where only the highly expressed HRG/pre-HRG subsets were analyzed. Looking at the HRG/pre-HRG expression ratios among highly expressed genes across all sample types could provide a clearer picture of any tissue and cell-type specific biases.

Figure 5: The authors should double check the statistical analysis in 5B. Validating the significance of HRG enrichment in sSPTs versus SPTs is important.

Figure 6: The authors compared the morphology of drone testicular tissue and the distribution of different sperm cell species between 3d and 14d, revealing a lack of early spermatogenic processes and an accumulation of late spermatid cells in 14d testes. However, sSPT cells did not exist in the testicular tissue at 14 days, and we could not observe the development track of sSPT cells. It is better to select the testicular tissue of drones in the intermediate period (example: 7d) for analysis, so that the comparison with 3d data can more effectively observe the changes and functions that may be involved in sSPT cells.

Figure 7: The authors chose 3d and 14d drones to prepare scRNA-seq libraries from freshly dissected testes. Then identified different cell types in the testicular tissue of the 3d drones, and suggesting a continuous process of spermatogenesis presented in the newly emerged honeybee testes. The article points out that 3d of age represents a transition stage when a massive number of spermatozoa are transferred from testes to seminal vesicles. However, from the results of Figure 1B, it seems that a 4-day drone is more appropriate.

Minor:

* Results p9: change ' aSPC-SPT ' for 'the aSPC-sSPT branch'

* Figure 6F: The "15 d testes" in the illustration does not correspond to the figure legends.

Reviewer #3: review pasted below (but also as an attachment for better formatting)

Overview:

This manuscript provides a detailed mapping of spermatogenesis at the single-cell level between recently emerged (3d) and sexually mature (14d) honeybee drones. The authors provide a well-characterised dataset using Drosophila marker genes to annotate cell types and identify a small subset of small spermatids in the newly emerged drones. They also describe ongoing sperm production post-emergence shown by the presence of cell-types and markers characteristic of the germline stem-cell niche and increasing spermatid numbers. This work will be a useful data source to better understand spermatogenesis in the honeybee and other Hymenoptera species by providing species-specific gene markers for the stages of male gamete production.

The annotation of clusters needs to be more clearly supported e.g. a more detailed explanation of the marker genes used with citations to show the appropriateness of said genes. Additionally, figure legends need to be more detailed; for example, abbreviations explained etc.

A few minor suggestions below:

Introduction

Paragraph 2: typo ‘development of honeybee’ and ‘in fruitfly’…. missing ‘the’?

Paragraph 3: Most of this paragraph might be better placed in results/discussion.

Results:

Testis scRNA-seq reveals a continuous process of spermatogenesis in newly-emerged honeybee drones

Paragraph 1:

‘visually inspected to be healthy’ could be more clear as to examples of what might be excluded.

States that 3d and 14d old drones were dissected and proceeds to describe data collected from drones aged 1-18 days in Fig 1C.

Paragraph 2:

‘showed testicular expression’ – explain what this means

‘corresponding bulk RNA-seq datasets’ - where are these described?

Paragraph 3:

‘Among these genes, 109 had enriched expression in at least one single-cell cluster of the 3d dataset

(Figure 1E; Materials and Methods)’ – Needs more information on thresholds for defining ‘enriched’ genes, even in the methods.

‘iteratively removed genes with either inconsistency or high ambiguity’ - any particular thresholds set for removal?

Why are late spermatogonia/spermatocytes abbreviated to aSPG and aSPC?

‘LOC409668 (Dmel\\PI31), LOC727432 (Dmel\\I-2), LOC100577491 (Dmel\\smn) and LOC414041 (Dmel\\Unr)’ – References to show significance of these genes? The annotation of the clusters needs to be more clearly supported.

The earliest spermatogonial cells express germline stem cell maintenance genes

Paragraph 1:

‘However, in the daily measurements of the sperm numbers, we detected a slight increase of the testicular spermatozoa at day 2 after emergence’ – statistically significant?

‘During days 4-8, the gain rate in SVs was greater than the loss rate in testes by 0.7~2.0 million per day (Figure S2)’ – statistically significant?

Paragraph 2:

‘top enrichment in functional categories such as…’ – More info on gene ontology analysis either here or in methods. E.g. what exact test and correction for adjusted p-values?

Paragraph 3:

‘eSPG1 > eSPG4 > eSPG2 > eSPG0 > eSPG3’ – rename according to order?

‘known GSC markers in Dmel, including nanos, LOC409301 (Dmel\\Mad), LOC409713 (Dmel\\hts), LOC412256 (Dmel\\E(bx)), LOC725878 (Dmel\\how), LOC727235 (Dmel\\ena) and LOC100576993 (Dmel\\zfh1)’ – References?

Spermatogenesis in honeybee is characterized by a branch point after meiosis II

Paragraph 1:

‘the aSPC-SPT branch appeared to uniquely represent a meiotic cell fate in honeybee’ – Typo? aSPC-sSPT?

‘Both groups showed a dynamic expression pattern with a peak level at the branch point, suggesting an active event of meiotic division II at this time (Figure 3B)’ – peak not particularly clear.. especially for G2/M genes. Not sure that the data fully supports this claim.

Hymenoptera-restricted genes are highly expressed in small spermatids

Paragraph 1:

‘the unique sSPT branch may be reflective of transcriptional programming events led by phylogeny specific genes in honeybee’ – what is the logic behind this hypothesis? Citations for context?

‘we examined the orthology relationship between Amel and Dmel in genes with sSPT-high expression (Group #2 in Figure 3D). We found that 44.5% of Group #2 genes share orthology with Dmel genes, and the proportion is significantly lower than any other group of genes’ – How does one conclude this from figure 3D? Should reference 4B here?

Transcription regulatory network uncovers transcription factors that contribute to small spermatid specification

Paragraph 1:

‘from our full-length RNA-seq data, scored motif clusters of annotated transcription factors for each TSS, and selected regulatory edges with significant weights and top-scored target genes to build the final regulons for SPTs and sSPTs (Materials and Methods)’ – not clear what exactly was done here, what was the scoring system?

How was the ‘connectivity’ mentioned in Figure 5C measured?

Paragraph 2:

‘In the sSPT network, 5 out of the 139 active TFs were significantly enriched HRGs as the relevant targets’ – Unclear, rephrase.

‘and sSPT-specific expression (Figure 5F)’ – Foxp and LOC100576369 also show significant increase in sSPT?

Spermatogenesis is largely absent in 14-day-old adult honeybee testes

Paragraph 2:

‘A riverplot shows the assignments of each integration cluster to the two datasets, revealing a lack of early spermatogenic processes and an accumulation of late spermatid cells in 14d testes (Figure 6E)’ – Confusing plot. 14d only data has not been introduced?

‘This finding was further verified by the bulk mRNA-seq detection of a general decrease in the maker genes for early spermatogenic cell types, including eSPG, aSPG, eSPC and aSPC (Figure 6F)’ – This plot shows 3d vs 15d when 3d vs 14d is what was being discussed prior.

Mutation load is decreased in spermatids of 14-day-old adult honeybee testes

Paragraph 1:

Typo: ‘3d and 14 testes’

‘By contrast, sSPTs had a higher per-base rate than either eSPTs or aSPTs, and this difference was greater than twofold’ – giving exact difference would be better.

Figures:

Figure 1D: May be clearer to label clusters with inferred cell type for consistency with text and panels 1G-H

Figure 4: Account for multiple testing with adjusted p-values?

Figure S3: Define arrow heads etc in legend.

**Have all data underlying the figures and results presented in the manuscript been provided?**

Reviewer #1: Yes

Reviewer #2: Yes

Reviewer #3: Yes

PLOS authors have the option to publish the peer review history of their article (what does this mean?). If published, this will include your full peer review and any attached files.

Reviewer #1: No

Reviewer #2: No

Reviewer #3: No

---

## [Decision Letter · Decision Letter 1]

13 Nov 2023

Dear Dr He,

Thank you very much for submitting your Research Article entitled 'A postmeiotically bifurcated roadmap of honeybee spermatogenesis marked by phylogenetically restricted genes' to PLOS Genetics.

The manuscript was fully evaluated at the editorial level and by independent peer reviewers. The reviewers appreciated the attention to an important topic but identified some concerns that we ask you address in a revised manuscript.

We therefore ask you to modify the manuscript according to the review recommendations. Your revisions should address the specific points made by each reviewer.

Yours sincerely,

Takaaki Daimon

Academic Editor

PLOS Genetics

Gregory P. Copenhaver

Editor-in-Chief

PLOS Genetics

The submitted revised manuscript has been evaluated by the three original reviewers.

The authors did a nice job in revising the manuscript, and as you will see below, reviewers #2 and #3 are very positive.

Although reviewer #1 acknowledges that the dataset obtained in this study is very significant and intriguing, he/she still has two concerns that should be addressed.

Therefore, I'd like to ask the authors to revise the manuscript.

Please note that reviewer #3 has attached the Word file indicating the possible changes to the manuscript.

Finally, if the authors agree, it would be nice to acknowledge the three anonymous reviewers in the Acknowledgments section.

Reviewer's Responses to Questions

**Comments to the Authors:**

Reviewer #1: Reply to the revised Yin et al: A postmeiotically bifurcated roadmap of honeybee spermatogenesis marked by phylogenetically restricted genes

I appreciate the authors’ response. Working with a non-model organism like Amel poses major challenges, and the authors should be commended for attempting to overcome them.

The authors responded to my first critique by adding additional analyses and language changes. However, I still have some concerns. It is very possible, as the authors state in their reply, that Dmel and Amel orthologs share similar functions. However, that does not mean that their expression is restricted to the same cell types or states. Even within the Drosophila genus, orthologs can be expressed in different cell types/states in different species. Indeed, it’s commonly accepted that since evolution more frequently acts on regulatory regions than coding regions, orthologs can have the same or similar coding region (as demonstrated by the authors in their substitution analysis), but dissimilar expression regulation. For example, pigment differences between Drosophila species are often not due to changes in Yellow or Ebony genes themselves, but changes in the regulation of those genes that results in differential cell type expression (e.g., PMID 12372246). Therefore, it’s still very difficult to determine if the annotation, based on 1-to-1 orthology between Amel and Dmel, is valid: these orthologus genes could be expressed in different cell types despite their orthology. Nevertheless, the authors refer to the annotation of the spermatogenic cell types as “unambiguous” in the text. I would posit that without validation there is, in fact, considerable ambiguity regarding their annotation.

In response to my second critique, the authors are very convincing as to why “sSPT” annotated cells are not simply low-quality or poorly sequenced cells, as the read quality and single cell against the second potential confounding factor – that the cell cluster arose from fragmenting spermatid heads and tails. Both spermatid heads and tails individually can 1) have high and complex read counts, as both contain considerable inherited RNA, 2) robustly cluster separately, as the cells’ fragile break point at the neck is consistent, and 3) appear in pseudotime analysis as germline cells that are derived from spermatocytes, since RNAs in both the head and the tail are largely transcribed in spermatocytes. I agree that the enriched expression of Hymenoptera-restricted genes in “sSPTs” is curious, but this result alone does not rule out alternative explanations for this cluster. Perhaps the authors could show a Nomarski image of their dissociated cell prep with quantification of fragmented spermatids. Additionally, such an image would allow for observation and quantification of morphologically sSPTs vs. SPTs in their cell preps. If the rate of fragmentation is very low, and the presence and ratios of (morphologically) sSPTs vs. SPTs tracks with the number of cells present in those annotated clusters, I agree it is likely that the annotated sSPTs are a bona fide cell state. As it stands, though, a limitations paragraph at the end does not change the confident language used throughout the paper to refer to this ambiguous cluster, including a sentence just one paragraph prior stating that “Our data captures a unique type of cells, small spermatids, which exhibit a clear sign of arising from an uneven meiosis II”. I do not believe that that there is such a clear sign, nor that the dataset has obviously captured these cells.

Overall, I still maintain that this is a dataset worthy of publication, especially given that honeybees are so understudied relative to their interesting biology. Evidence of asymmetric meiosis would be highly impactful, and would be a hugely interesting result to everyone in the meiosis community if validated. I hope the authors can successfully validate their exciting hypotheses from this dataset.

Reviewer #2: No

Reviewer #3: I reviewed this article with a trusted postdoc. Both of us were happy with the changes and both commented that the manuscript is much more readable.

**Have all data underlying the figures and results presented in the manuscript been provided?**

Reviewer #1: None

Reviewer #2: None

Reviewer #3: Yes

PLOS authors have the option to publish the peer review history of their article (what does this mean?). If published, this will include your full peer review and any attached files.

Reviewer #1: No

Reviewer #2: No

Reviewer #3: No

---

## [Editor Report · Decision Letter 2]

22 Nov 2023

Dear Dr He,

We are pleased to inform you that your manuscript entitled "A postmeiotically bifurcated roadmap of honeybee spermatogenesis marked by phylogenetically restricted genes" has been editorially accepted for publication in PLOS Genetics. Congratulations!

Yours sincerely,

Takaaki Daimon

Academic Editor

PLOS Genetics

Gregory P. Copenhaver

Editor-in-Chief

PLOS Genetics

Comments from the Associate Editor:

The authors have nicely revised the manuscript according to the reviewers' suggestions.

I think it is now ready for publication.

Thank you very much for submitting this beautiful work to PLOS Genetics.

**Data Deposition**

http://datadryad.org/submit?journalID=pgenetics&manu=PGENETICS-D-23-00740R2

**Press Queries**

---

## [Editor Report · Acceptance letter]

27 Nov 2023

PGENETICS-D-23-00740R2 

A postmeiotically bifurcated roadmap of honeybee spermatogenesis marked by phylogenetically restricted genes 

Dear Dr He, 

We are pleased to inform you that your manuscript entitled "A postmeiotically bifurcated roadmap of honeybee spermatogenesis marked by phylogenetically restricted genes" has been formally accepted for publication in PLOS Genetics! Your manuscript is now with our production department and you will be notified of the publication date in due course.

With kind regards,

Katalin Szabo

PLOS Genetics

On behalf of:
